# TEST-TIME ADAPTATION WITHOUT SOURCE DATA FOR OUT-OF-DOMAIN BIOACTIVITY PREDICTION

**Yiming Yang**,[*] **Zhiyuan Zhou**,[*] **Yueming Yin,** **Hoi-Yeung Li**,[†] **Adams Wai-Kin Kong**[†]
Nanyang Technological University
{yiming014,zhou0457}@e.ntu.edu.sg,
{yueming.yin,hyli,adamskong}@ntu.edu.sg

## ABSTRACT

Accurate prediction of protein-ligand bioactivity is a cornerstone of modern drug discovery, yet current deep learning methods often struggle with out-of-domain (OOD) generalization. The existing solutions rely on access to source data, making them impractical in scenarios where data cannot be accessed due to confidentiality, privacy concerns or intellectual property restrictions. In this paper, we provide the first exploration of a more realistic setting for bioactivity prediction, where models are expected to adapt to distribution shifts without access to source data. Motivated by the critical role of binding-relevant interactions in determining ligand-protein bioactivity, we introduce an uncertainty-weighted consistency strategy, in which original samples with high confidence guide their augmented counterparts by minimizing feature distance. This encourages the model to focus on informative interaction regions while suppressing reliance on spurious or non-causal substructures. To further enhance representation discriminability and prevent feature collapse, we integrate a contrastive optimization objective that pulls together augmented views of the same complex and pushes away views from different complexes. Together, these two components enable the learning of invariant, bioactivity-aware representations, allowing robust adaptation under distribution shifts. Extensive experiments across DTIGN, SIU 0.6, and DrugOOD demonstrate that our framework consistently outperforms state-of-the-art baselines under scaffold, protein, and assay based OOD settings. Especially on the eight subsets of DTIGN, it improves Pearson's $R$ by 8.2% and Kendall's Tau $\tau$ by 5.8% on average over the best baseline, underscoring its effectiveness as a source data-absent solution for OOD bioactivity prediction.

## 1 INTRODUCTION

The traditional drug discovery process is limited by lengthy operation cycles and enormous economic costs, with the development of a single new drug typically requiring nearly a decade and about \$3 billion in investment (Hughes et al., 2011; Ejalonibu et al., 2021). Recent advances in deep learning have spurred growing interest in AI-driven drug discovery, enabling considerable breakthroughs in diverse areas such as molecular property prediction (Zitnick et al., 2022; Ying et al., 2021), protein folding (Jumper et al., 2021; Lin et al., 2023), and protein-ligand interaction modeling (Kong et al., 2024; Wang et al., 2023). These methods aim to accelerate the identification of promising ligands and elucidate complex biological mechanisms, thereby reducing the reliance on manual processes. In this work, we focus on the problem of bioactivity prediction, which seeks to identify ligand molecules capable of modulating the functional outcomes of target proteins.

With the increasing adoption of pocket-ligand modeling (Yin et al., 2024; Huang et al., 2025), graph neural networks (GNNs) have become particularly prominent (Zhang et al., 2023; Wang et al., 2024b; Mastropietro et al., 2023; Li et al., 2021), as graph-based representations offer a natural and powerful means to characterize molecular structures and interactions. Representative models such as DTIGN (Yin et al., 2024) and AttentionMGT-DTA (Wu et al., 2024a) exhibit impressive capabilities

---

[*]Equal contribution.
[†]Corresponding author.

across a variety of tasks (e.g., $IC_{50}$, $EC_{50}$, $K_d$, and $K_i$). Despite their potential, existing methods largely rely on the underlying hypothesis that training and testing data are independently sampled from an identical domain (Kao et al., 2021; Yuan et al., 2022; Wu et al., 2024a; Yang et al., 2023). In practice, however, real-world domains are often dynamic and uncertain, characterized by varying experimental conditions, novel molecular scaffolds, and previously unseen proteins (Ji et al., 2023; Huang et al., 2025), which require the model to effectively handle distribution shifts (Ji et al., 2023; Yin et al., 2021). Even worse, target domains usually have limited labeled data from experiments for training and fine-tuning. For instance, the COVID-19 pandemic highlights how an unpredictable event can introduce entirely new target proteins from an unknown distribution (Platto et al., 2020), underscoring the urgent need to enhance current methods with respect to OOD generalization.

Although approaches based on invariant learning (Sagawa et al., 2019; Wu et al., 2024b; Wang et al., 2024a) and graph-based generalization (Wu et al., 2022; Zhu et al., 2021) could be applied to mitigate the problem, they still face a key limitation in real applications: the requirement for full access to source data, which may not be feasible due to confidentiality, privacy concerns or intellectual property restrictions. To this end, we introduce a realistic but challenging source data-absent bioactivity prediction setting with only a well-trained source model provided as supervision. To address this challenging OOD setting for bioactivity prediction, we propose **TAB**—a **T**est-time **A**daptation framework for out-of-domain **B**ioactivity Prediction, which jointly leverages uncertainty-weighted consistency learning and contrastive learning for effective source data-absent adaptation.

Our motivation for TAB is two-fold, as illustrated in Figure 1. On the one hand, bioactivity is inherently determined by specific biological interactions within the pocket-ligand complex. A ligand cannot function independently, while its activity critically depends on both the target protein and its surrounding space. In this context, geometric features play a central role in representing binding status and have been proven highly effective in modeling molecular interactions (Wei et al., 2024; Han et al., 2024; Song et al., 2024; Zhang et al., 2022b). In particular, E(3)-equivariant GNNs capture local spatial relationships on protein surfaces and significantly improve binding-site prediction (Wei et al., 2024). As such, we aim to direct the model's attention toward the binding region and the spatial arrangement between interacting molecules. On the other hand, bioactivity prediction models are susceptible to *privileged substructure bias*, where certain ligand moieties or recurrent protein motifs frequently occur in active complexes but are neither causal determinants of binding nor bioactiv-

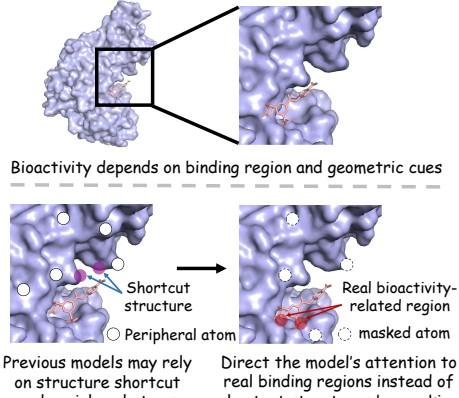

Figure 1: Motivation of TAB. The protein and ligand are shown in purple and brown, respectively.

ity. Such spurious correlations can be exploited as shortcut signals, leading to overfitting and poor generalization. For example, kinase inhibitor and GPCR ligand datasets contain many active ligands with methyl-substituted phenyl rings (Klekota & Roth, 2008; Schneider & Schneider, 2017); likewise, proteins may exhibit recurring surface patterns correlated with activity but not mechanistically responsible for binding (Johansson et al., 2013). In both cases, models risk attributing activity to non-causal substructures, thereby limiting their robustness across domains.

Building on these insights, we first introduce the consistency component, where the augmented samples are guided to align with their original confident counterparts, thereby discouraging reliance on spurious shortcuts and promoting bioactivity-relevant invariant representations. This design naturally guides the model's attention toward informative binding regions, where geometric cues and interaction features are most critical for bioactivity (Zhou et al., 2025). In addition, to account for varying levels of prediction reliability across samples, we incorporate an uncertainty-weighted regularization that prioritizes more confident predictions and stabilizes adaptation under challenging OOD conditions, as shown in Figure 2.

Beyond consistency learning, we integrate a contrastive component to enhance representation discriminability and prevent feature collapse. By encouraging alignment between augmented views of the same complex while separating representations of different complexes, this design reinforces binding-relevant signals and sharpens the distinction between activity-related and irrelevant features.

Combined with consistency learning, it yields complementary benefits that enable robust generalization across diverse OOD settings under realistic, source data-absent conditions.

In summary, our contributions can be listed as follows:

- To the best of our knowledge, no existing work has studied bioactivity prediction under OOD settings without access to source data. In this work, we introduce a realistic, yet challenging source data-absent bioactivity prediction setting in which only a well-trained source model is available for supervision.

- Inspired by the importance of binding-relevant interactions and the need to avoid spurious shortcuts, we propose a self-supervised framework that integrates uncertainty-weighted consistency learning with contrastive optimization. This design directs the model's attention to informative pocket-ligand interactions, suppresses reliance on false privileged substructures, and promotes the learning of invariant, bioactivity-aware representations.

- Through extensive experiments on the DTIGN, SIU 0.6, and DrugOOD benchmarks, we demonstrate that TAB achieves state-of-the-art performance under scaffold, protein, and assay based OOD settings, delivering improvements in both predictive accuracy and ranking consistency. On the eight subsets of DTIGN, TAB improves Pearson's $R$ by 8.2% and Kendall's Tau $\tau$ by 5.8% on average over the best baseline, underscoring its effectiveness as a source data-absent solution for OOD bioactivity prediction.

## 2 RELATED WORK

**Bioactivity Prediction.** Accurately predicting the bioactivity of ligands against proteins represents a pivotal step in drug discovery. Early methods such as DeepDTA (Öztürk et al., 2018) and MT-DTI (Shin et al., 2019) operate directly on ligand SMILES strings and protein sequences using CNN or attention-based encoders. Subsequently, GraphDTA (Nguyen et al., 2021) represents drugs as molecular graphs, which enables the model to directly capture bonds among atoms. Furthermore, GIGN (Yang et al., 2023) and DTIGN (Yin et al., 2024) construct geometric interaction graphs that integrate both ligand and protein information, allowing the model to better capture complex binding patterns. By progressively incorporating structural information, this class of methods provides a more faithful representation of molecular interactions and thus has become the predominant paradigm in recent bioactivity prediction research.

**Out-of-Domain Generalization in Bioactivity Prediction.** The OOD issue poses a critical challenge in bioactivity prediction. Previous methods in alleviating this problem can be roughly categorized into three groups: invariant learning, data augmentation, and graph-based approaches. Approaches in the first group (Sagawa et al., 2019; Yin et al., 2021; 2022; Wang et al., 2024a; Wu et al., 2024b) aim to capture the invariant patterns across different training environments. Data augmentation (Wang et al., 2021) is able to help OOD generalization as it can enrich the training distribution. The last group methods (Wu et al., 2022; Zhu et al., 2021) tackle this problem by exploiting graph structural information and identifying subgraphs that remain predictive across different domains. However, they all require full access to source data, either to construct multiple training environments for learning invariance, to generate augmented samples from the original distribution, or to analyze graph structures in search of transferable subgraphs. In contrast, we propose to mitigate OOD effects at test time to address the demand from real applications, where source data is not available. This setting has never been studied according to the best of our knowledge.

**Test-Time Adaptation in CV.** Test-Time Adaptation (TTA) has been widely studied in conventional Computer Vision (CV) tasks for improving robustness. Approaches include self-supervised tasks such as rotation prediction (TTT (Sun et al., 2020)) and masked image generation (TTT-MAE (Gandelsman et al., 2022)), as well as pseudo-labeling methods like SHOT (Liang et al., 2020) and MEMO (Zhang et al., 2022a). Despite these advances, their potential for bioactivity prediction remains unexplored, where molecular data pose unique challenges due to structural complexity, inherent substructure biases, and binding region modeling (Johansson et al., 2013; Han et al., 2024). Motivated by this gap, we develop the first TTA method tailored to bioactivity prediction.

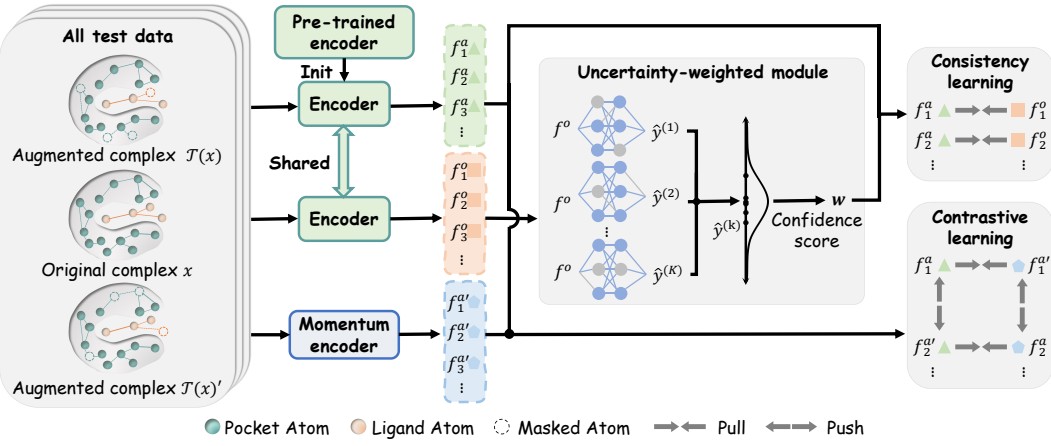

Figure 2: The framework of TAB.

# 3 METHOD

## 3.1 PROBLEM FORMULATION

We consider the task of source data-absent bioactivity prediction under domain shifts. Each input sample $x \in \mathcal{G}$ is a pocket-ligand complex graph represented as $x = (V, E)$, where $V$ is the set of nodes corresponding to atoms, and $E$ is the set of edges representing chemical bonds. Each node encodes atomic and chemical features, while edges capture both covalent and non-covalent bonds.

In this setting, only a pretrained source model is available as supervision, and direct access to the source dataset is prohibited, including raw samples, intermediate features, and statistics. At test time, we assume access to only the unlabeled target samples $\mathcal{D}_{test} = \{(x_i)\}_{i=1}^{N_{test}}, x_i \in \mathcal{G}$. Specifically, the model parameters are updated by minimizing a self-supervised objective $\mathcal{L}_{ssl}$. Formally, the adaptation process can be written as:

$$\min_{\theta} \mathbb{E}_{x \sim \mathcal{D}_{test}} \left[ \mathcal{L}_{ssl}(f_\theta(x)) \right], \tag{1}$$

where $f_\theta$ denotes the prediction model with parameters $\theta$. The comparisons between our setting and existing adaptation settings in the bio-domain are summarized in Table 1. While existing settings employed in previous bio-related studies (Li et al., 2023; Moon et al., 2023; He et al., 2022; Zeng et al., 2024; Liu et al., 2023) serve for specific purposes, they fail to fully cover all practical scenarios where source data, target data, or supervision are not simultaneously available.

Table 1: Characteristics of problem settings in previous bio-related studies for adapting a trained model to a potentially shifted test domain. Among the source and target data $(x^s, x^t)$ and labels $(y^s, y^t)$, our test-time setting only requires access to the target data $x^t$.

| Setting | Source data | Target data | Training loss | Testing loss |
|---|---|---|---|---|
| Fine-tuning (Li et al., 2023; Moon et al., 2023) | ✗ | $x^t, y^t$ | $\mathcal{L}(x^t, y^t)$ | - |
| Continual learning (He et al., 2022) | ✗ | $x^t, y^t$ | $\mathcal{L}(x^t, y^t)$ | - |
| Domain adaptation (Liu et al., 2023; Zeng et al., 2024) | $x^s, y^s$ | $x^t$ | $\mathcal{L}(x^s, y^s) + \mathcal{L}(x^s, x^t)$ | - |
| Test-time adaptation (ours) | ✗ | $x^t$ | ✗ | $\mathcal{L}(x^t)$ |

## 3.2 UNCERTAINTY-WEIGHTED CONSISTENCY LEARNING

Taking inspiration from the critical role of binding-relevant interactions, we adopt a consistency objective to encourage the model to focus on informative pocket-ligand regions while suppressing reliance on spurious privileged substructures. The key idea is to align the representations of each pocket-ligand complex with those of its perturbed counterpart, thereby emphasizing bioactivity-relevant invariant features, as depicted in the upper part of Figure 2.

Formally, let $\mathbf{f}_i^o = f_\theta(x_i)$ and $\mathbf{f}_i^a = f_\theta(\mathcal{T}(x_i))$ denote the feature representations of the original and augmented samples, respectively, where $\mathcal{T}(\cdot)$ denotes the perturbation process. For each pocket-

---

**Algorithm 1:** TAB framework

---

**Input:** target dataset $D_{test}$, online encoder $f_\theta$, momentum encoder $f_{\theta'}$, memory queue $\mathcal{Q}$, momentum $m$, number of Monte Carlo dropout passes $K$, temperature $\tau_c$, consistency weight $\alpha$, contrastive weight $\beta$

**Output:** Labels of samples in target dataset $D_{test}$

---

1   **Function** `UncerWeight`$(x^o)$:
2     **for** $k = 1$ **to** $K$ **do**
3       $\mathbf{f}^{(k)} \leftarrow f_\theta(x^o)$
4     $\mu \leftarrow \frac{1}{K}\sum_{k=1}^{K}\mathbf{f}^{(k)}$
5     $\sigma^2 \leftarrow \frac{1}{K-1}\sum_{k=1}^{K}\|\mathbf{f}^{(k)}-\mu\|^2$
6     $w \leftarrow 1/(\sigma^2+\epsilon)$
7     **return** $w$

8   **Function** `ConsLoss`$(x^o, x^a, w)$:
9     $\mathbf{f}^o \leftarrow f_\theta(x^o), \quad \mathbf{f}^a \leftarrow f_\theta(x^a)$
10     $\mathcal{L}_{cons} \leftarrow \mathbb{E}[\, w \cdot (1 - \cos(\mathbf{f}^o, \mathbf{f}^a))\,]$
11     **return** $\mathcal{L}_{cons}$

12   **Function** `ContrLoss`$(x^a, x^{a'}, \mathcal{Q}, \tau_c)$:
13     $\mathbf{f}^a \leftarrow f_\theta(x^a), \quad \mathbf{f}^{a'} \leftarrow f_{\theta'}(x^{a'})$

14     # no gradient on $\mathbf{f}^{a'}$
15     $\mathcal{L}_{ctr} \leftarrow \mathbb{E}[\, \exp(\mathbf{f}_i^a \cdot \mathbf{f}_i^{a'}/\tau_c)/ (\sum_{j=1}^{B}\exp(\mathbf{f}_i^a \cdot \mathbf{f}_j^{a'}/\tau_c) + \sum_{q=1}^{|\mathcal{Q}|}\exp(\mathbf{f}_i^a \cdot \mathbf{f}_q^a/\tau_c))\,]$
16     **return** $\mathcal{L}_{ctr}$

17   **Main procedure:**
18     **foreach** $x \in D_{test}$ **do**
19       $x^o \leftarrow x, \quad x^a \leftarrow \mathcal{T}(x)$
20       $w \leftarrow$ `UncerWeight`$(x^o)$
21       $\mathcal{L}_{cons} \leftarrow$ `ConsLoss`$(x^o, x^a, w)$
22       $x^{a'} \leftarrow \mathcal{T}(x)'$
23       $\mathcal{L}_{ctr} \leftarrow$ `ContrLoss`$(x^a, x^{a'}, \mathcal{Q}, \tau_c)$
24       $\mathcal{L}_{ssl} \leftarrow \alpha \cdot \mathcal{L}_{cons} + \beta \cdot \mathcal{L}_{ctr}$
25       Update $\theta$ by gradient descent using $\mathcal{L}_{ssl}$
26       $\theta' \leftarrow m \cdot \theta' + (1-m) \cdot \theta$

---

ligand complex graph, we randomly mask a subset of atom and edge features. These augmentations tend to obscure false privileged substructures, thereby suppressing reliance on spurious shortcuts and encouraging the extraction of bioactivity-relevant invariant representations. Since the interface accounts for only a small portion of the entire complex, the majority of the masked content corresponds to atoms and edges outside the binding site, which in turn naturally directs attention toward the binding region. Even if some key atoms and edges are temporarily occluded, essential geometric cues and binding poses remain largely intact. We then enforce consistency by minimizing the cosine distance between $\mathbf{f}_i^o$ and $\mathbf{f}_i^a$, encouraging the model to suppress reliance on spurious privileged substructures while maintaining sensitivity to binding-relevant interactions.

Moreover, we note that not all samples are equally reliable. To reduce the influence of noisy or uncertain predictions, we introduce an uncertainty-weighted regularization scheme. Specifically, we estimate confidence via Monte Carlo dropout: performing $K$ stochastic forward passes with the original source model on the original input yields multiple feature samples $f_\theta^{(k)}(x_i)$, from which the mean $\mu_i$ and variance $\sigma_i^2$ are computed as:

$$\mu_i = \frac{1}{K}\sum_{k=1}^{K} f_\theta^{(k)}(x_i), \quad \sigma_i^2 = \frac{1}{K-1}\sum_{k=1}^{K}\|f_\theta^{(k)}(x_i) - \mu_i\|^2. \tag{2}$$

Accordingly, the confidence score for a given sample is defined as $w_i = 1/(\sigma_i^2 + \epsilon)$, with $\epsilon$ serving as a stabilizing constant. Thus, samples with higher confidence are assigned heavier weights during training, enabling the model to prioritize trustworthy information while mitigating the risk of error propagation. Finally, the overall consistency training objective is defined as:

$$\mathcal{L}_{cons} = \frac{1}{B}\sum_{i=1}^{B} w_i \cdot \left(1 - \frac{\mathbf{f}_i^o \cdot \mathbf{f}_i^a}{\|\mathbf{f}_i^o\|\,\|\mathbf{f}_i^a\|}\right), \tag{3}$$

where $B$ is the mini-batch size.

### 3.3   JOINT CONTRASTIVE LEARNING

While consistency learning guides the model to focus on binding-relevant regions, it does not explicitly encourage discrimination between complexes. To address this, we integrate a contrastive objective that promotes feature separability. Following the shared instance-discrimination principle (Chen et al., 2020; He et al., 2020), we treat features of two augmented views of the same

pocket-ligand complex as positive pairs, while features from different complexes serve as negative pairs. Specifically, given a target input $x$, we reuse the augmented view $\mathcal{T}(x)$ already employed in the consistency objective, and generate one additional augmented view $\mathcal{T}(x)'$, as shown in the bottom panel of Figure 2. This design ensures that the consistency-guided view reinforces alignment around binding-relevant signals, while the second augmentation introduces complementary variations that enrich the representation.

To increase the diversity of negative samples, we maintain a memory queue $\mathcal{Q}$, which stores feature representations from previous mini-batches. The update is done by enqueue and dequeue in a FIFO manner. To stabilize the feature space and prevent rapid fluctuations, we adopt a MoCo-style (He et al., 2020) momentum encoder $f_{\theta'}(\cdot)$, whose parameters $\theta'$ are initialized from the pretrained source weights and updated via exponential moving average:

$$\theta' \leftarrow m\theta' + (1-m)\theta, \tag{4}$$

where $m \in [0,1)$ is the momentum coefficient and $\theta$ denotes the parameters of the online encoder. This momentum update allows the encoder to provide stable feature representations for contrastive learning without relying on backpropagation for every mini-batch. Specifically, we encode the consistency-guided augmentation $\mathcal{T}(x)$ and the additional augmentation $\mathcal{T}(x)'$ into embeddings $\mathbf{f}^a = f_\theta(\mathcal{T}(x))$ and $\mathbf{f}^{a'} = f_{\theta'}(\mathcal{T}'(x))$, with the momentum embedding $\mathbf{f}^{a'}$ enqueued into $\mathcal{Q}$. The contrastive loss encourages the online embedding $\mathbf{f}^a$ to be close to its positive counterpart $\mathbf{f}^{a'}$, while pushing it away from all negative embeddings in the current mini-batch and the memory queue $\mathcal{Q}$:

$$\mathcal{L}_{ctr} = -\frac{1}{B}\sum_{i=1}^{B}\log\frac{\exp(\mathbf{f}_i^a \cdot \mathbf{f}_i^{a'}/\tau_c)}{\sum_{j=1}^{B}\exp(\mathbf{f}_i^a \cdot \mathbf{f}_j^{a'}/\tau_c) + \sum_{q=1}^{|\mathcal{Q}|}\exp(\mathbf{f}_i^a \cdot \mathbf{f}_q^a/\tau_c),} \tag{5}$$

where $B$ is the mini-batch size, $\mathbf{f}_i^a$ and $\mathbf{f}_i^{a'}$ are the online and momentum embeddings of the $i$-th sample, $\mathbf{f}_j^{a'}$ indexes other in-batch embeddings, $\mathbf{f}_q^a$ runs over embeddings stored in the memory queue $\mathcal{Q}$, and $\tau_c > 0$ is the temperature controlling the sharpness of the similarity distribution.

Finally, the overall loss for TAB combines the uncertainty-weighted consistency loss and the contrastive loss: $\mathcal{L}_{ssl} = \alpha\mathcal{L}_{cons} + \beta\mathcal{L}_{ctr}$. The algorithm of TAB is given in Algo 1.

## 4 EXPERIMENTS

### 4.1 DATASETS AND METRICS

**DTIGN.** We first adopt the DTIGN dataset (Yin et al., 2024), which focuses on scaffold-based OOD evaluation. Training and test sets are explicitly split according to the molecular scaffolds of small molecules, ensuring that test molecules have distinct scaffolds from those seen during training. Specifically, DTIGN comprises 8 protein targets (I1, I2, I3, I4, I5, E1, E2, and E3). Within each subset, hundreds to thousands of ligands are associated with the corresponding protein and target label $IC_{50}$ or $EC_{50}$.

**SIU 0.6.** We then employ the SIU 0.6 dataset (Huang et al., 2025), which is designed for protein-based OOD evaluation. A fixed test set is provided, and to ensure OOD generalization, any proteins in the training set with sequence identity greater than 60% to the test set are removed. Specifically, for the $K_i$ task, there are 33,595 training samples and 3,153 test samples, corresponding to 195 training proteins and 10 test proteins. For the $K_d$ task, there are 12,333 training samples and 1,723 test samples, corresponding to 161 training proteins and 7 test proteins.

**DrugOOD.** Finally, we evaluate DrugOOD (Ji et al., 2023) on the structure-based affinity prediction dataset. For the assay subset, each domain corresponds to a distinct experimental assay. The training set contains 55 such domains with 5,541 samples, while the test set includes 111 domains with 3,047 samples. For the protein subset, each domain represents a unique protein. The training set consists of 29 proteins with 5,524 samples, and the test set comprises 66 proteins with 3,006 samples. Notably, there is no overlap domain between the training and test sets.

The original dataset is simplified into a binary classification task by discretizing continuous bioactivity measurements. We restore it to its natural regression form, which better preserves the continuous signals and maintains sensitivity to subtle yet biologically meaningful variations. In addition, since

Table 2: Comparison of RMSE, Pearson's $R$, and Kendall's Tau $\tau$ results for different methods on DTIGN across multiple tasks. The best performance is highlighted in bold.

| Metric | Method | Dataset | | | | | | | | Avg |
|---|---|---|---|---|---|---|---|---|---|---|
| | | I1 | I2 | I3 | I4 | I5 | E1 | E2 | E3 | |
| RMSE ↓ | ERM | 1.282 | 1.672 | 1.474 | **1.167** | 1.125 | 0.855 | 0.864 | 1.231 | 1.209 |
| | IRM | 1.478 | 1.873 | 1.744 | 1.290 | 1.150 | 0.918 | 0.830 | 1.265 | 1.318 |
| | GroupDro | 1.447 | 1.644 | 1.675 | 1.268 | 1.232 | 0.885 | 0.837 | 1.324 | 1.289 |
| | CIA-LRA | 1.317 | 1.621 | 1.684 | 1.221 | 1.239 | 0.909 | 0.834 | 1.353 | 1.272 |
| | CaNet | 1.573 | 1.682 | 1.609 | 1.180 | 1.144 | 0.935 | 0.855 | 1.373 | 1.294 |
| | DANN | 1.387 | 1.746 | 1.738 | 1.238 | 1.169 | 0.949 | 0.805 | 1.295 | 1.291 |
| | AFSE | 1.461 | 1.712 | 1.665 | 1.497 | 1.348 | 0.963 | 0.919 | 1.287 | 1.357 |
| | Mixup-GNN | 1.435 | 1.773 | 1.820 | 1.461 | 1.177 | 0.970 | 0.973 | 1.354 | 1.370 |
| | EERM | 1.326 | 1.540 | 1.576 | 1.440 | 1.232 | 0.863 | 0.854 | 1.281 | 1.264 |
| | SR-GNN | 1.277 | 1.595 | 1.681 | 1.227 | 1.167 | 0.874 | 0.832 | 1.352 | 1.251 |
| | TAB | **1.180** | **1.481** | **1.433** | 1.217 | **1.121** | **0.841** | **0.778** | **1.207** | **1.157** |
| R ↑ | ERM | 0.592 | 0.533 | 0.474 | 0.389 | 0.349 | 0.200 | 0.462 | 0.316 | 0.414 |
| | IRM | 0.306 | 0.465 | 0.343 | 0.320 | 0.305 | -0.033 | 0.355 | 0.236 | 0.287 |
| | GroupDro | 0.332 | 0.464 | 0.332 | 0.298 | 0.256 | 0.145 | 0.376 | 0.258 | 0.308 |
| | CIA-LRA | 0.427 | 0.438 | 0.285 | 0.361 | 0.260 | 0.196 | 0.409 | 0.286 | 0.314 |
| | CaNet | 0.427 | 0.443 | 0.314 | 0.293 | 0.298 | 0.200 | 0.440 | 0.191 | 0.326 |
| | DANN | 0.478 | 0.476 | 0.507 | 0.257 | 0.311 | 0.088 | 0.425 | 0.257 | 0.350 |
| | AFSE | 0.546 | 0.373 | 0.382 | 0.302 | 0.316 | 0.133 | 0.415 | 0.220 | 0.336 |
| | Mixup-GNN | 0.308 | 0.395 | 0.296 | 0.193 | 0.255 | 0.035 | 0.109 | 0.110 | 0.213 |
| | EERM | 0.469 | 0.559 | 0.352 | 0.277 | 0.302 | 0.194 | 0.386 | 0.262 | 0.350 |
| | SR-GNN | 0.482 | 0.480 | 0.351 | 0.342 | 0.332 | 0.209 | 0.422 | 0.286 | 0.363 |
| | TAB | **0.614** | **0.605** | **0.537** | **0.401** | **0.359** | **0.216** | **0.507** | **0.349** | **0.448** |
| τ ↑ | ERM | 0.417 | 0.393 | 0.314 | 0.273 | 0.226 | 0.139 | 0.330 | 0.271 | 0.295 |
| | IRM | 0.223 | 0.306 | 0.237 | 0.232 | 0.216 | 0.005 | 0.250 | 0.204 | 0.209 |
| | GroupDro | 0.238 | 0.306 | 0.226 | 0.218 | 0.167 | 0.083 | 0.274 | 0.227 | 0.217 |
| | CIA-LRA | 0.310 | 0.322 | 0.194 | 0.258 | 0.178 | 0.120 | 0.308 | 0.245 | 0.234 |
| | CaNet | 0.301 | 0.321 | 0.175 | 0.206 | 0.203 | 0.115 | 0.320 | 0.138 | 0.223 |
| | DANN | 0.353 | 0.317 | 0.349 | 0.178 | 0.218 | 0.090 | 0.308 | 0.232 | 0.256 |
| | AFSE | 0.375 | 0.270 | 0.249 | 0.211 | 0.221 | 0.069 | 0.293 | 0.208 | 0.237 |
| | Mixup-GNN | 0.217 | 0.276 | 0.197 | 0.201 | 0.168 | 0.020 | 0.100 | 0.132 | 0.164 |
| | EERM | 0.331 | 0.410 | 0.232 | 0.158 | 0.197 | 0.136 | 0.282 | 0.248 | 0.250 |
| | SR-GNN | 0.370 | 0.348 | 0.235 | 0.238 | 0.227 | **0.160** | 0.308 | 0.256 | 0.268 |
| | TAB | **0.427** | **0.419** | **0.356** | **0.279** | **0.245** | 0.138 | **0.350** | **0.282** | **0.312** |

DrugOOD only provides ligand SMILES and protein amino acid sequences, the interaction information between ligands and targets cannot be directly modeled. To address this limitation, we perform molecular docking. Detailed procedures for docking are provided in Appendix C.

**Metrics.** To ensure fair comparison with previous works, we adopt widely used evaluation metrics in bioactivity prediction, including root mean square error (RMSE), Pearson correlation coefficient $R$, Kendall's Tau correlation coefficient $\tau$. Following the original SIU 0.6 paper, we additionally report mean absolute error (MAE) and Spearman correlation coefficient ($\rho$) for it.

## 4.2 EXPERIMENTAL SETTINGS

We compare TAB with several baseline approaches. The standard empirical risk minimization (ERM) paradigm is included as a basic reference. In addition, we evaluate invariant learning methods IRM (Arjovsky et al., 2019), GroupDro (Sagawa et al., 2019), CIA-LRA (Wang et al., 2024a) and CaNet (Wu et al., 2024b); domain generalization methods DANN (Ganin et al., 2016) and AFSE (Yin et al., 2022); graph data augmentation methods Mixup-GNN (Wang et al., 2021); and graph OOD methods EERM (Wu et al., 2022) and SR-GNN (Zhu et al., 2021), all of which require training with access to source data. In contrast, TAB operates in a source-absent setting. For a fair comparison, we adopt DTIGN (Yin et al., 2024) as the backbone model across all experiments. Additional details and hyperparameter settings are provided in Appendix E.

Table 3: Comparison of MAE, Pearson's $R$, Kendall's Tau $\tau$, and Spearman's $\rho$ results across different methods on SIU 0.6 for $K_d$ and $K_i$ tasks. The best performance is highlighted in bold.

| Dataset | Method | Metric | | | |
|---|---|---|---|---|---|
| | | MAE↓ | R↑ | τ↑ | ρ↑ |
| $K_d$ | ERM | 1.095 | 0.320 | 0.215 | 0.319 |
| | IRM | - | - | - | - |
| | GroupDro | - | - | - | - |
| | CIA-LRA | 1.175 | 0.174 | 0.125 | 0.184 |
| | CaNet | 1.171 | 0.245 | 0.185 | 0.271 |
| | DANN | 1.203 | 0.025 | 0.038 | 0.060 |
| | AFSE | 1.069 | 0.304 | 0.247 | 0.374 |
| | Mixup-GNN | 1.100 | 0.361 | 0.253 | 0.386 |
| | EERM | 1.284 | 0.098 | 0.033 | 0.052 |
| | SR-GNN | **1.028** | 0.384 | 0.257 | 0.381 |
| | TAB | 1.055 | **0.393** | **0.283** | **0.419** |
| $K_i$ | ERM | 1.653 | 0.123 | 0.060 | 0.091 |
| | IRM | - | - | - | - |
| | GroupDro | - | - | - | - |
| | CIA-LRA | 1.679 | 0.100 | 0.036 | 0.055 |
| | CaNet | 1.610 | 0.034 | 0.029 | 0.043 |
| | DANN | 1.727 | 0.060 | 0.074 | 0.110 |
| | AFSE | 1.652 | 0.089 | 0.070 | 0.104 |
| | Mixup-GNN | 1.600 | 0.091 | 0.043 | 0.065 |
| | EERM | 1.805 | -0.027 | 0.026 | 0.040 |
| | SR-GNN | 1.612 | 0.003 | -0.011 | -0.015 |
| | TAB | **1.415** | **0.141** | **0.115** | **0.175** |

Table 4: Comparison of RMSE, Pearson's $R$, and Kendall's Tau $\tau$ across different methods on DrugOOD. The best performance is highlighted in bold.

| Dataset | Method | Metric | | |
|---|---|---|---|---|
| | | RMSE↓ | R↑ | τ↑ |
| Assay | ERM | 1.506 | 0.119 | 0.063 |
| | IRM | 1.541 | 0.229 | 0.142 |
| | GroupDro | 1.666 | 0.152 | 0.010 |
| | CIA-LRA | 1.487 | 0.265 | 0.170 |
| | CaNet | **1.437** | 0.269 | 0.153 |
| | DANN | 1.612 | -0.061 | -0.032 |
| | AFSE | 1.519 | -0.027 | -0.005 |
| | Mixup-GNN | 1.539 | -0.015 | -0.004 |
| | EERM | 1.525 | 0.236 | 0.111 |
| | SR-GNN | 1.502 | 0.242 | 0.145 |
| | TAB | 1.552 | **0.388** | **0.230** |
| Protein | ERM | 1.367 | 0.018 | 0.032 |
| | IRM | 1.483 | 0.092 | 0.061 |
| | GroupDro | 1.459 | 0.021 | 0.015 |
| | CIA-LRA | 1.492 | -0.020 | -0.030 |
| | CaNet | 1.454 | 0.032 | 0.027 |
| | DANN | 1.327 | 0.065 | 0.042 |
| | AFSE | 1.351 | -0.032 | -0.005 |
| | Mixup-GNN | 1.578 | 0.009 | -0.021 |
| | EERM | 1.369 | 0.110 | 0.076 |
| | SR-GNN | 1.536 | 0.051 | 0.036 |
| | TAB | **1.319** | **0.144** | **0.073** |

## 4.3 EXPERIMENTAL RESULTS

We first compare TAB with existing methods under the scaffold OOD setting on the DTIGN dataset. As shown in Table 2, the average performance reveals that existing OOD approaches fail to deliver gains over standard ERM. In contrast, TAB achieves superior results across nearly all evaluation metrics. Specifically, TAB reduces RMSE by 4.3%, increases Pearson's $R$ by 8.2%, and improves Kendall's Tau $\tau$ by 5.8% on average compared to the best baseline. These findings demonstrate that TAB not only lowers prediction error but also substantially enhances correlation-based measures, yielding more reliable rankings of pocket-ligand interactions. Importantly, the improvements are consistent across diverse protein targets, highlighting the robustness and generalizability of our method in the scaffold OOD setting.

Besides, we evaluate TAB on SIU 0.6 dataset under the protein OOD setting. As reported in Table 3, TAB consistently outperforms alternatives on both tasks. We note that IRM and GroupDro are not applicable due to the lack of domain partitioning in the SIU 0.6 dataset, and thus their results are denoted as '-'. For $K_d$ task, TAB achieves the highest Pearson's $R$ (0.393), Kendall's Tau $\tau$ (0.283), and Spearman's $\rho$ (0.419), outperforming the strongest baseline by a clear margin while maintaining competitive MAE. Similarly, for the $K_i$ task, TAB delivers substantial improvements, raising Pearson's $R$ from 0.123 to 0.141, Kendall's Tau $\tau$ from 0.060 to 0.115, and Spearman's $\rho$ from 0.091 to 0.175. These consistent gains across multiple correlation-based metrics confirm that TAB more accurately captures the relative binding affinities, thereby yielding more reliable predictions in OOD scenarios.

Finally, we present the results on DrugOOD in Table 4, where TAB achieves the best overall performance across nearly all metrics. On the assay task, while TAB shows a slightly higher RMSE (1.552) compared to ERM (1.506), it substantially improves the correlation metrics, increasing Pearson's $R$ from 0.119 to 0.388 and Kendall's Tau $\tau$ from 0.063 to 0.230. On the protein task, TAB delivers consistent gains, lowering RMSE from 1.367 to 1.319 (a 3.5% reduction) and boosting Pearson's $R$ from 0.018 to 0.144 and Tau $\tau$ from 0.032 to 0.073. Together with the scaffold and protein OOD gains observed on DTIGN and SIU 0.6, these results highlight TAB's robustness and effectiveness across diverse types of domain shift.

### 4.4 IN-DEPTH ANALYSIS OF TAB

#### 4.4.1 ABLATION STUDY

In our proposed TAB, we incorporate both uncertainty-weighted consistency learning and contrastive learning to obtain more robust predictions. To assess the contribution of each component, we conduct ablation experiments on the DTIGN dataset, with results summarized in Table 5.

Table 5: Ablation study on DTIGN dataset. ''*w/o contr*'' and ''*w/o cons*'' denote removing the contrastive learning and consistency learning modules, respectively. The best performance is highlighted in bold.

| Dataset | TAB | | | TAB *w/o contr* | | | TAB *w/o cons* | | | ERM | | |
|---|---|---|---|---|---|---|---|---|---|---|---|---|
| | RMSE↓ | R↑ | τ↑ | RMSE↓ | R↑ | τ↑ | RMSE↓ | R↑ | τ↑ | RMSE↓ | R↑ | τ↑ |
| I1 | **1.180** | **0.614** | 0.427 | 1.221 | 0.594 | 0.420 | 1.265 | 0.610 | **0.429** | 1.282 | 0.592 | 0.417 |
| I2 | **1.481** | **0.605** | **0.419** | 1.493 | **0.605** | 0.369 | 1.503 | 0.559 | 0.399 | 1.672 | 0.533 | 0.393 |
| I3 | **1.433** | **0.537** | **0.356** | 1.526 | 0.488 | 0.317 | 1.441 | 0.482 | 0.303 | 1.474 | 0.474 | 0.314 |
| I4 | 1.217 | **0.401** | 0.279 | 1.247 | 0.395 | **0.280** | 1.258 | 0.297 | 0.211 | **1.167** | 0.389 | 0.273 |
| I5 | **1.121** | 0.359 | **0.245** | 1.174 | 0.325 | 0.207 | 1.154 | 0.358 | 0.240 | 1.125 | 0.349 | 0.226 |
| E1 | **0.841** | 0.216 | 0.138 | 0.894 | 0.196 | 0.126 | 0.923 | 0.202 | **0.143** | 0.855 | 0.200 | 0.139 |
| E2 | **0.778** | **0.507** | **0.350** | 0.785 | 0.421 | 0.275 | 0.926 | 0.482 | 0.340 | 0.864 | 0.462 | 0.330 |
| E3 | **1.207** | **0.349** | **0.282** | 1.260 | 0.319 | 0.274 | 1.273 | 0.286 | 0.263 | 1.231 | 0.316 | 0.271 |
| Avg | **1.157** | **0.448** | **0.312** | 1.191 | 0.432 | 0.285 | 1.201 | 0.427 | 0.295 | 1.209 | 0.414 | 0.295 |

We can see that each component positively contributes to the overall performance, with the full model TAB consistently achieving the best results compared with its two variants, TAB *w/o contr* and TAB *w/o cons*. Interestingly, we also observe that incorporating either consistency or contrastive learning in isolation may occasionally underperform the baseline. This phenomenon is not surprising: applying only consistency learning can lead to overly compressed representations, thereby weakening discriminability, while applying only contrastive learning may amplify spurious differences without proper regularization. When combined, the two modules act in a complementary manner, promoting the learning of invariant yet informative representations that more faithfully capture the underlying bioactivity signals.

#### 4.4.2 CASE STUDY ON BINDING REGION ATTRIBUTION

We employ a perturbation-based attribution strategy to assess whether TAB effectively focuses on the true binding region. For each pocket-ligand complex, we randomly remove a subset of ligand atoms, their connected pocket atoms, and the associated inter-molecular edges, thereby simulating scenarios where key binding site information is missing. The model predictions on the original and perturbed inputs are denoted as $\hat{y}_{\text{ori}}$ and $\hat{y}_{\text{per}}$, and their difference is defined as $\Delta\hat{y} = \hat{y}_{\text{ori}} - \hat{y}_{\text{per}}$. Larger $\Delta\hat{y}$ indicates greater sensitivity to critical binding sites, which is desirable. We report the average value $\overline{\Delta\hat{y}}$ across all sampled subsets in the DTIGN dataset, as shown in Figure 3. Since atom and edge removal disrupts essential binding interactions, the bioactivity is expected to decrease, and thus $\Delta\hat{y}$ should generally be positive. Interestingly, negative $\Delta\hat{y}$ are observed in the ERM baseline, highlighting its flawed reliance on irrelevant sites. By comparison, TAB consistently yields significantly higher $\overline{\Delta\hat{y}}$, demonstrating its stronger and more reliable focus on the relevant interaction regions.

Figure 3: Prediction variance of TAB and ERM baseline under perturbation-based attribution.

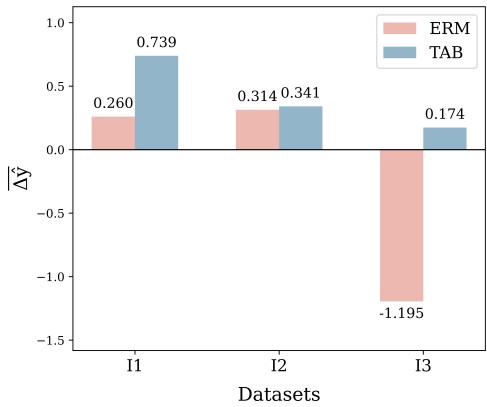

## 5 CONCLUSION

In this paper, we address the research gap concerning source data-absent out-of-domain (OOD) bioactivity prediction. We propose TAB, a test-time adaptation framework that captures molecular interaction-aware features by focusing on bioactivity-relevant regions. By combining uncertainty-weighted consistency learning with contrastive optimization, TAB suppresses spurious correlations while enhancing feature discriminability, enabling robust and bioactivity-aware representations. Extensive experiments demonstrate its effectiveness across diverse domain shifts, providing a practical source-absent solution for real-world bioactivity prediction.

### ACKNOWLEDGMENTS

We gratefully acknowledge Nanyang Biologics Pte. Ltd. (NYB) for providing travel support to attend the conference.

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

# A ADDITIONAL EXPERIMENTAL ANALYSIS

## A.1 SENSITIVITY ANALYSIS OF LOSS WEIGHTS

We conduct a sensitivity analysis on the consistency weight $\alpha$ and the contrastive weight $\beta$ to examine the robustness of TAB. To ensure fair comparability across different settings, we fix $\alpha + \beta = 2$ and evaluate a wide range of $\alpha/\beta$ configurations. The detailed results are reported in Table 6.

Table 6: Sensitivity analysis of the consistency weight $\alpha$ and contrastive weight $\beta$.

| Dataset | Method | $\alpha$ | $\beta$ | RMSE↓ | R↑ | $\tau$↑ |
|---------|--------|----------|---------|-------|------|---------|
| I1 | ERM | – | – | 1.282 | 0.592 | 0.417 |
|  | TAB | 0.2 | 1.8 | 1.265 | 0.590 | 0.424 |
|  | TAB | 0.6 | 1.4 | 1.374 | 0.602 | 0.424 |
|  | TAB | 1.0 | 1.0 | 1.180 | 0.614 | 0.427 |
|  | TAB | 1.4 | 0.6 | 1.165 | 0.614 | 0.417 |
|  | TAB | 1.8 | 0.2 | 1.289 | 0.605 | 0.426 |
| I2 | ERM | – | – | 1.672 | 0.533 | 0.393 |
|  | TAB | 0.2 | 1.8 | 1.460 | 0.603 | 0.420 |
|  | TAB | 0.6 | 1.4 | 1.453 | 0.601 | 0.427 |
|  | TAB | 1.0 | 1.0 | 1.481 | 0.605 | 0.419 |
|  | TAB | 1.4 | 0.6 | 1.408 | 0.597 | 0.439 |
|  | TAB | 1.8 | 0.2 | 1.427 | 0.599 | 0.411 |
| I3 | ERM | – | – | 1.474 | 0.474 | 0.314 |
|  | TAB | 0.2 | 1.8 | 1.448 | 0.463 | 0.327 |
|  | TAB | 0.6 | 1.4 | 1.458 | 0.458 | 0.319 |
|  | TAB | 1.0 | 1.0 | 1.433 | 0.537 | 0.356 |
|  | TAB | 1.4 | 0.6 | 1.552 | 0.498 | 0.340 |
|  | TAB | 1.8 | 0.2 | 1.588 | 0.510 | 0.347 |

Overall, TAB exhibits stable performance across a broad range of $\alpha/\beta$ ratios, suggesting that the proposed framework is not overly sensitive to the exact weighting between consistency regularization and contrastive learning. In particular, the default setting $\alpha = 1$ and $\beta = 1$ consistently achieves competitive or near-optimal results across all datasets, validating our choice of equal weighting in the main experiments.

## A.2 Effect of uncertainty weighting

We further conduct an ablation study to assess the impact of uncertainty-aware weighting. Specifically, we replace the uncertainty-weighted consistency loss with uniform weighting, referred to as *TAB w/o uw*. As shown in Table 7, removing uncertainty weighting leads to consistent performance degradation across all datasets. This observation confirms that uncertainty-aware weighting plays a critical role in TAB, as it adaptively down-weights unreliable predictions during test-time optimization, thereby reducing the risk of propagating noisy or misleading updates.

Table 7: Ablation study on uncertainty-weighting.

| Dataset | Method | RMSE↓ | R↑ | τ ↑ |
|---|---|---|---|---|
| I1 | ERM | 1.282 | 0.592 | 0.417 |
| | TAB w/o uw | 1.232 | 0.591 | 0.422 |
| | TAB | 1.180 | 0.614 | 0.427 |
| I2 | ERM | 1.672 | 0.533 | 0.393 |
| | TAB w/o uw | 1.609 | 0.576 | 0.416 |
| | TAB | 1.481 | 0.605 | 0.419 |
| I3 | ERM | 1.474 | 0.474 | 0.314 |
| | TAB w/o uw | 1.435 | 0.484 | 0.336 |
| | TAB | 1.433 | 0.537 | 0.356 |

## A.3 Effect of EMA and masking strategies

We conduct ablation studies to analyze the impact of the EMA update strategy and the proposed masking mechanism. Specifically, *TAB w/o EMA* removes the EMA-based parameter update, *TAB (substitution)* replaces atom masking with random atom substitution, *TAB (coord. jitter)* applies random perturbations to atomic 3D coordinates instead of masking, and *TAB (mask)* corresponds to our original design. The results are summarized in Table 8.

Table 8: Effect of EMA update and masking strategies.

| Dataset | Method | RMSE↓ | R↑ | τ ↑ |
|---|---|---|---|---|
| I1 | ERM | 1.282 | 0.592 | 0.417 |
| | TAB w/o EMA | 1.272 | 0.572 | 0.418 |
| | TAB (substitution) | 1.232 | 0.592 | 0.412 |
| | TAB (coord. jitter) | 1.470 | 0.503 | 0.347 |
| | **TAB (mask)** | **1.180** | **0.614** | **0.427** |
| I2 | ERM | 1.672 | 0.533 | 0.393 |
| | TAB w/o EMA | 1.585 | 0.556 | 0.404 |
| | TAB (substitution) | 1.513 | 0.581 | 0.416 |
| | TAB (coord. jitter) | 1.665 | 0.543 | 0.398 |
| | **TAB (mask)** | **1.481** | **0.605** | **0.419** |
| I3 | ERM | 1.474 | 0.474 | 0.314 |
| | TAB w/o EMA | 1.467 | 0.474 | 0.323 |
| | TAB (substitution) | 1.449 | 0.477 | 0.319 |
| | TAB (coord. jitter) | 1.493 | 0.446 | 0.299 |
| | **TAB (mask)** | **1.433** | **0.537** | **0.356** |

## A.4 Effect of batch size

We investigate the impact of batch size by evaluating four different settings (4, 8, 16, and 32). Across all datasets, TAB consistently outperforms the pretrained ERM baseline, demonstrating strong ro-

bustness to batch size variations as shown in Table 9. Notably, we observe a rise-then-fall trend as the batch size increases: moderate batch sizes generally yield the best performance, while overly large batches lead to slightly diminished gains.

Table 9: Effect of batch size.

| Dataset | Method | Batch Size | RMSE↓ | R↑ | $\tau$ ↑ |
|---------|--------|-----------|-------|------|------|
| I1 | ERM | – | 1.282 | 0.592 | 0.417 |
|    | TAB | 4 | 1.186 | 0.604 | 0.442 |
|    | TAB | 8 | 1.166 | 0.627 | 0.448 |
|    | TAB | 16 | 1.180 | 0.614 | 0.427 |
|    | TAB | 32 | 1.272 | 0.601 | 0.431 |
| I2 | ERM | – | 1.672 | 0.533 | 0.393 |
|    | TAB | 4 | 1.641 | 0.550 | 0.392 |
|    | TAB | 8 | 1.453 | 0.581 | 0.409 |
|    | TAB | 16 | 1.481 | 0.605 | 0.419 |
|    | TAB | 32 | 1.508 | 0.566 | 0.396 |
| I3 | ERM | – | 1.474 | 0.474 | 0.314 |
|    | TAB | 4 | 1.443 | 0.487 | 0.328 |
|    | TAB | 8 | 1.438 | 0.483 | 0.334 |
|    | TAB | 16 | 1.433 | 0.537 | 0.356 |
|    | TAB | 32 | 1.446 | 0.490 | 0.400 |

## B DETAILS OF BASELINE METHODS

**IRM** (Arjovsky et al., 2019)
Invariant Risk Minimization (IRM) is a learning paradigm developed to tackle the poor out-of-distribution (OOD) generalization of traditional Empirical Risk Minimization (ERM); ERM relies on the independent and identically distributed (IID) assumption, which often leads it to absorb spurious correlations from training data (e.g., cows being disproportionately associated with green pastures and camels with deserts in image classification) and thus fail when test data distribution shifts, whereas IRM leverages multiple distinct training environments (e.g., different data collection locations, times, or experimental conditions) to learn invariant causal correlations instead of environment-specific spurious ones. It aims to find a data representation such that the optimal classifier built on this representation is consistent across all training environments, and is formulated as a constrained optimization problem that balances two goals: minimizing predictive error on training data (aligning with ERM) and imposing a penalty to enforce the invariance of the classifier across environments, thereby enhancing models' ability to generalize to unseen test distributions by prioritizing invariant causal relationships over spurious ones.

**GroupDro** (Sagawa et al., 2019)
Group Distributionally Robust Optimization (GroupDRO) is a group-level variant of Distributionally Robust Optimization (DRO) designed to address the limitation that overparameterized neural networks often achieve high average accuracy but fail on atypical data groups (e.g., due to spurious correlations). It minimizes the worst-case training loss across pre-defined data groups, enabling robustness to group shifts. Critically, naive GroupDRO underperforms in overparameterized regimes; thus, it requires coupling with strong regularization penalties to reduce group-specific generalization gaps and boost worst-group test accuracy while preserving high average accuracy. Additionally, an efficient stochastic optimization algorithm with convergence guarantees is proposed to scale Group-DRO to large models and datasets.

**CIA** (Wang et al., 2024a)
To tackle the limitation that traditional invariant learning methods (e.g., IRM) fail in node-level graph OOD generalization due to insufficient class-conditional invariance constraints, Cross-environment Intra-class Alignment (CIA) is proposed. CIA eliminates spurious features by aligning representations of same-class nodes from different environments—these nodes share similar invari-

ant causal patterns but differ in spurious features. By enforcing this class-conditional invariance (lacking in IRM), CIA accurately identifies invariant causal patterns. For node-level OOD scenarios where environment labels are unavailable, CIA-LRA (Localized Reweighting Alignment) is further developed. It avoids environment labels via three key designs: 1) aligning only local node pairs (2–6 hops) to prevent invariant feature collapse; 2) reweighting pairs based on neighborhood label distributions; 3) integrating an invariant subgraph extractor for refined aggregation. It is theoretically guaranteed by a PAC-Bayesian OOD error bound.

**CaNet** (Wu et al., 2024b)
Causal Intervention for Network Data (CaNet) is to tackle the OOD generalization issue of GNNs, where GNNs suffer performance degradation due to latent environmental confounding bias (leading to over-reliance on environment-sensitive spurious correlations between ego-graph features and node labels). Without prior environment labels, CaNet integrates an environment estimator (inferring pseudo-environments from ego-graphs) and a mixture-of-expert GNN predictor (dynamically selecting propagation branches via pseudo-environments). Its causal inference-derived objective mitigates confounding bias and learns environment-invariant relations.

**DANN** (Ganin et al., 2016)
Domain-Adversarial Neural Networks (DANN) is a domain adaptation approach for neural networks, aiming to learn features that are discriminative for the source-domain task yet invariant to source-target domain shifts. It enhances feed-forward architectures with a gradient reversal layer (GRL) and jointly optimizes three components: a feature extractor, a label predictor (for source-task classification), and a domain classifier (to distinguish source/target domains). The feature extractor is trained to minimize the source label loss while maximizing the domain classifier loss (via GRL reversing gradients during backpropagation), enabling adaptation with labeled source data and unlabeled target data (no target labels needed).

**AFSE** (Yin et al., 2022)
Adversarial Feature Subspace Enhancement (AFSE) is proposed to tackle the poor generalization of deep graph learning (DGL) models in predicting GPCR-targeting ligand bioactivities, a problem caused by small-sized/biased training datasets and inherent activity cliffs. It dynamically generates rich representations in new feature subspaces through bidirectional adversarial learning, and minimizes the maximum loss of molecular divergence and bioactivity to ensure local smoothness of model outputs.

**Mixup-GNN** (Wang et al., 2021)
This work introduces Mixup strategies tailored for both node and graph classification tasks. For node classification, a two-branch Mixup graph convolution is employed to combine paired nodes' receptive-field subgraphs. To mitigate interference, a two-stage framework is adopted: the first stage obtains neighbor representations through a standard GNN, while the second stage applies Mixup convolutions using these representations. For graph classification, the method interpolates graph-level representations directly in the semantic space, which helps reduce overfitting in GNNs.

**EERM** (Wu et al., 2022)
Explore-to-Extrapolate Risk Minimization (EERM) targets OOD generalization for node-level graph tasks. It employs multiple adversarial context generators (graph structure editors) to maximize risk variance across virtual environments while minimizing mean risk, enabling GNNs to extrapolate from a single observed environment. The method has theoretical guarantees for valid OOD solutions and shows effectiveness in handling shifts from artificial spurious features, cross-domain transfers, and dynamic graph evolution across diverse GNN backbones.

**SR-GNN** (Zhu et al., 2021)
Shift-Robust GNN (SR-GNN) is designed to tackle distributional shift from biased training data in GNN-based semi-supervised learning—a critical issue overlooked by most existing GNNs. As a general framework, it works for both deep traditional GNNs (e.g., GCN) and shallow linearized GNNs: for deep models, it uses Central Moment Discrepancy (CMD) regularization to reduce representation shift between biased training and unbiased IID samples; for linearized models, it employs instance reweighting to improve training sample representativeness.

## C    DETAILS OF DATASETS

**DTIGN dataset** (Yin et al., 2024)
The dataset of DTIGN is specifically designed to support a significant task: improving bioactivity prediction by integrating drug-target interaction information, which addresses the limitation of traditional datasets that only focus on ligand structures. It forms a unique benchmark to enable interaction-aware bioactivity prediction evaluation. The bioactivity data is derived from the ChEMBL database, screened to target single proteins with over 1000 bioactive small molecules (resulting in 900 target proteins from 6778 human proteins in ChEMBL), and preprocessed to obtain 6.75 million valid records covering metrics like $IC_{50}$ and $EC_{50}$. The DTIGN dataset contains eight protein targets (denoted as I1, I2, I3, I4, I5, E1, E2, and E3), each treated as an independent subset. Within each subset, the ligand scaffolds are diverse, and the test set was manually constructed to evaluate the ability to identify ligands with promising bioactivity. Consequently, the dataset is inherently non-I.I.D. The information of DTIGN is shown in Table 10.

Table 10: Summary statistics of the DTIGN dataset

| Subset | Task | #Data points |
|--------|------|--------------|
| I1 | $IC_{50}$ | 926 |
| I2 | $IC_{50}$ | 1,509 |
| I3 | $IC_{50}$ | 3,501 |
| I4 | $IC_{50}$ | 5,920 |
| I5 | $IC_{50}$ | 9,336 |
| E1 | $EC_{50}$ | 996 |
| E2 | $EC_{50}$ | 1,639 |
| E3 | $EC_{50}$ | 3,532 |

**SIU dataset** (Huang et al., 2025)
The SIU dataset is a recently introduced benchmark for bioactivity prediction that provides multiple types of bioactivity measurements. Two versions are available: SIU-0.9 and SIU-0.6, where the number denotes the sequence similarity threshold applied to remove test samples that are overly similar to those in the training set. In this study, we use SIU-0.6, since the sequence would be much more diverse, and we choose $K_d$ and $K_i$ as the target label. The information of DTIGN is shown in Table 11.

Table 11: Summary statistics of the SIU subsets

| Dataset | Task | Subset | #Data points |
|---------|------|--------|--------------|
| SIU-0.6 | $K_i$ | train | 33,595 |
|  |  | valid | 3,787 |
|  |  | test | 3,153 |
|  | $K_d$ | train | 12,333 |
|  |  | valid | 1,422 |
|  |  | test | 1,723 |

**DrugOOD dataset** (Ji et al., 2023)
DrugOOD is a systematic out-of-distribution (OOD) dataset curator and benchmark tailored for AI-aided drug discovery (AIDD), with a specific focus on the critical task of drug-target binding affinity prediction involving both small-molecule compounds and macromolecular protein targets. For the present study, we utilized subsets from structure-based affinity prediction (SBAP) task of DrugOOD, specifically focusing on datasets under the $EC_{50}$ measurement and core noise level. Detailed statistics of the utilized subsets are shown in the Table 12.

It should be noted that the original DrugOOD dataset for the SBAP task lacks structural information for proteins and ligands, providing only protein sequences and ligand SMILES. For each protein,

we retrieve its PDB ID using the UniProt identifier and obtain the corresponding PDB structure. For each ligand, we generate ligand 3D structures by converting SMILES using the RDKit toolkit. Then, potential binding sites are identified by FPocket (Le Guilloux et al., 2009) to define the docking box. Finally, we generate docking pocket-ligand complexes with Vina-GPU 2.1 (Tang et al., 2024).

Table 12: Summary statistics of the DrugOOD subsets

| Dataset | #Training | #Validation | #Testing | #Training Env | #Validation Env | #Test Env | #Label |
|---|---|---|---|---|---|---|---|
| EC50-assay-core | 6,387 | 3,562 | 3,527 | 51 | 60 | 122 | $EC_{50}$ |
| EC50-protein-core | 6,420 | 3,544 | 3,505 | 29 | 28 | 72 | $EC_{50}$ |

## D  DETAILS OF METRICS

**RMSE**
Root Mean Squared Error (RMSE) is a widely used metric for evaluating regression models. It measures the square root of the average squared difference between predicted values and actual values. RMSE penalizes larger errors more strongly, making it sensitive to outliers. A lower RMSE indicates a better fit of the model to the data.

$$\text{RMSE} = \sqrt{\frac{1}{n} \sum_{i=1}^{n} (y_i - \hat{y}_i)^2}$$

where $n$ is the number of samples, $y_i$ is the true value, and $\hat{y}_i$ is the predicted value.

**Pearson Correlation Coefficient**
The Pearson Correlation Coefficient ($R$) is a statistical measure that evaluates the linear relationship between two variables. Its value ranges from $-1$ to $+1$: a value close to $+1$ indicates a strong positive correlation, a value close to $-1$ indicates a strong negative correlation, and a value near $0$ suggests little or no linear correlation.

$$r = \frac{\sum_{i=1}^{n} (x_i - \bar{x}) (y_i - \bar{y})}{\sqrt{\sum_{i=1}^{n} (x_i - \bar{x})^2} \sqrt{\sum_{i=1}^{n} (y_i - \bar{y})^2}}$$

where $x_i$ and $y_i$ are the observed values of the two variables, $\bar{x}$ and $\bar{y}$ are the means of $x$ and $y$, $n$ is the number of samples.

**Kendall's Tau Correlation Coefficient**
Kendall's Tau ($\tau$) is a non-parametric measure of the ordinal association between two variables. It evaluates the similarity of the orderings of data when ranked by each variable. The coefficient ranges between $-1$ and $+1$, where a value close to $+1$ indicates strong agreement, $-1$ indicates strong disagreement, and $0$ suggests no association.

$$\tau = \frac{C - D}{\frac{1}{2} n(n - 1)}$$

where $C$ is the number of concordant pairs, $D$ is the number of discordant pairs, and $n$ is the total number of data points.

**MAE**
Mean Absolute Error (MAE) is a common metric for evaluating regression models. It measures the average of the absolute differences between predicted values and actual values. Unlike RMSE, MAE treats all errors with equal weight, making it less sensitive to outliers. A lower MAE indicates better model performance.

$$\text{MAE} = \frac{1}{n} \sum_{i=1}^{n} |y_i - \hat{y}_i|$$

where $n$ is the number of samples, $y_i$ is the true value, and $\hat{y}_i$ is the predicted value.

**Spearman's Rank Correlation Coefficient**
Spearman's Rank Correlation Coefficient ($\rho$) is a non-parametric measure of the monotonic relationship between two variables. Unlike Pearson correlation, which captures linear dependence, Spearman correlation is based on the ranked values of the data. Its value ranges from $-1$ (perfect negative correlation) to $+1$ (perfect positive correlation), with $0$ indicating no correlation.

$$\rho = 1 - \frac{6 \sum_{i=1}^{n} d_i^2}{n\left(n^2 - 1\right)}$$

where $d_i = R(x_i) - R(y_i)$ is the difference between the ranks of $x_i$ and $y_i$, and $n$ is the total number of samples.

## E    IMPLEMENTATION DETAILS

For the uncertainty-weighted regularization, we pre-store a database of dropout-based outputs from the original source model with the original inputs to improve computational efficiency. Specifically, we apply a dropout rate of 0.1 and perform five forward passes to compute the standard deviation. For contrastive learning, the momentum of the key encoder is 0.999, and the temperature for the contrastive loss is 0.1. We set both loss weights for consistency learning and contrastive learning to $\alpha = \beta = 1.0$ across all experiments. Model training is conducted using the AdamW optimizer with a StepLR scheduler. All experiments are implemented in PyTorch with Python 3.8 and executed on an Ubuntu server equipped with four NVIDIA GeForce RTX 4090 GPUs.

## F    LIMITATIONS

A key limitation of TAB is its reliance on pocket-ligand complex information. When experimentally resolved structures are unavailable, docking simulations must be performed to generate conformations, adding computational overhead. Nevertheless, with GPU acceleration, docking time remains acceptable. Developing methods that can directly leverage sequence information without complex structure modeling remains an important direction for future work.

