# OpenReview forum: "Test-Time Adaptation without Source Data for Out-of-Domain Bioactivity Prediction"
_ICLR.cc/2026/Conference — ICLR 2026 Poster_

### Official Review · Reviewer_UabT · 2025-10-27

**Soundness:** 1
**Presentation:** 3
**Contribution:** 2
**Rating:** 2
**Confidence:** 4

**Summary:**

The paper presents a method for test time adaptation for OOD protein-ligand binding prediction. The method consists in finetuning the encoder of a GNN architecture with two objectives: consistency (a transformation is applied to the input and the method learns to project augmented versions of the same input into the same embedding space, the loss is weighted by the internal certainty of the model, to minimize noise due to transformations that may deform the input) and contrastive learning (the method learns to project close together augmented views of the same input and further apart from augmented views of other inputs).

**Strengths:**

1. The method is interesting. Its application domain, when no source data exists apart from the features at inference time is highly relevant and captures realistic scenarios.
2. The evaluation datasets are relevant and baselines are comprehensive.

**Weaknesses:**

1. Title is misleading. I think that protein-ligand binding affinity is a more accurate description of the method rather than bioactivity. Some bioactivities like antibacterial, anticancer, allergenic, do not necessarily involve explicit protein-ligand interactions and this method is limited to modelling those types of interactions.
2. Results should contain a measure of the dispersion of the results across multiple runs, otherwise it is impossible to determine the significance of the differences.
3. Along the lines of 2, the contribution of the contrastive learning, or the consistency learning are difficult to determine. For example, from the baseline model (ERM) to TAB w/o cons there is a difference of 0.008 in RMSE, which does not seem significant, suggesting that the contrastive learning is not providing that much information. Similarly TAB w/o contr has a performance of 1.191 which is only 0.034 from the complete method. Looking at the results in Table 2, it seems like the standard deviation across datasets is higher than those improvements, indicating that there is not enough evidence to make any claims regarding the benefits of TAB.

**Questions:**

Is there any specific reason that I may be missing as to why the title focuses on bioactivity rather than binding affinity prediction?

---

> ### Author Response · Authors · 2025-11-21
> **Response to Weakness 1**
>
> Thank you for the insightful and constructive feedback. We are especially grateful for your acknowledgement of the practical significance of our "source-free" setting, which requires no access to source data during inference. Below we address each point in turn.
>
> ### In response to Weakness 1
>
> We would like to respectfully clarify the source of the confusion.
>
> First, the definition of bioactivity prediction used in our paper follows the scope established in recent literature. In *"Redefining the Task of Bioactivity Prediction"* (ICLR 2025) [1], the authors formally revisit the concept of bioactivity prediction and introduce the SIU dataset. They explicitly state that:
> - "In this context, **'bioactivity’ encompasses the diverse biological effects resulting from small molecule–protein interactions**, including binding responses—commonly quantified by Kd and Ki—as well as functional responses, typically assessed through IC50 and EC50."
> - "We find that the key issue stems from the improper definition of the current bioactivity prediction task, particularly in terms of both data construction and evaluation metrics."
> - "To address these issues, **we propose redefining the bioactivity prediction task by constructing a large-scale structural dataset of small molecule–protein interactions**, featuring multiple small molecules for each protein target."
>
> **Our work adheres exactly to this formulation**: we study **protein-target–specific bioactivity prediction** and evaluate on SIU [1] and DTIGN [2], both purpose-built for modeling ligand bioactivity toward protein targets.
>
> Secondly, we fully acknowledge the nuanced distinction between "binding affinity" and "bioactivity". Binding affinity quantifies the strength of a physical interaction, whereas bioactivity, as defined in [1], captures the biological consequence of that interaction. A molecule may bind tightly but exhibit weak biological response (e.g., binding to a non-functional or allosteric site). We would like to model these functional responses rather than affinity alone, hence "bioactivity prediction" is the accurate terminology within this setting.
>
> Third, we note that the underlying mechanisms of antibacterial, anticancer, and allergenic bioactivities are generally mediated through specific protein-ligand interactions. For example, anticancer drugs exert therapeutic effects by binding to cancer-related protein targets such as EGFR, FGFR3, and HER2; antibacterial agents act by inhibiting essential bacterial proteins (e.g., PBPs and DNA gyrase); and allergenic responses arise through protein-mediated immune signaling. Therefore, even when the experimental readout is phenotypic—such as cell viability or immune activation—the determinants of these bioactivities are still rooted in molecular interactions with protein targets. In principle, our method can be extended to these domains, as it is designed to model protein-target–specific bioactivity, consistent with the definition formalized in recent literature.
>
> [1] Redefining the Task of Bioactivity Prediction, ICLR, 2025
>
> [2] Advancing bioactivity prediction through molecular docking and self-attention, IEEE Journal of Biomedical and Health Informatics, 2024

---

> > ### Author Response · Authors · 2025-11-21
> > **Response to Weakness 2 and 3**
> >
> > ### In response to Weakness 2
> >
> > For all datasets, we have conducted five independent runs and now report both the mean and standard deviation of our methods. The updated results are provided below, showing that our method achieves consistently superior performance with low variance across runs, demonstrating its robustness and statistical reliability.
> >
> > **Results on DTIGN across Multiple Runs**
> >
> > |Dataset|Method|RMSE(↓)|Pearson(↑)|Tau(↑)|
> > |:-:|:-:|:-:|:-:|:-:|
> > |**I1**|ERM|1.282|0.592|0.417|
> > ||TAB|**1.191**±0.059|**0.610**±0.015|**0.437**±0.013|
> > ||
> > |**I2**|ERM|1.672|0.533|0.393|
> > ||TAB|**1.421**±0.052|**0.613**±0.013|**0.427**±0.008|
> > |**I3**|ERM|1.474|0.474|0.314|
> > ||TAB|**1.446**±0.015|**0.488**±0.036|**0.335**±0.016|
> > ||
> > |**I4**|ERM|1.167|0.389|0.273|
> > ||TAB|1.221±0.008|**0.396**±0.005|**0.276**±0.004|
> > ||
> > |**I5**|ERM|1.125|0.349|0.226|
> > ||TAB|1.127±0.007|**0.352**±0.005|**0.233**±0.006|
> > ||
> > |**E1**|ERM|0.855|0.200|0.139|
> > ||TAB|**0.808**±0.030|**0.226**±0.025|**0.156**±0.036|
> > ||
> > |**E2**|ERM|0.864|0.462|0.330|
> > ||TAB|**0.796**±0.016|**0.483**±0.017|**0.338**±0.006|
> > ||
> > |**E3**|ERM|1.231|0.316|0.271|
> > ||TAB|**1.227**±0.012|**0.342**±0.016|**0.287**±0.004|
> > ||
> > |**Avg**|ERM|1.209|0.414|0.295|
> > ||TAB|**1.155**±0.025|**0.439**±0.017|**0.311**±0.012|
> >
> > **Results on SIU 0.6 across Multiple Runs**
> >
> > |Dataset|Method|MAE(↓)|Pearson(↑)|Tau(↑)|Spearman(↑)|
> > |:-:|:-:|:-:|:-:|:-:|:-:|
> > |**Kd**|ERM|1.095|0.320|0.215|0.319|
> > ||TAB|**1.065**±0.010|**0.396**±0.006|**0.277**±0.006|**0.410**±0.007|
> > ||
> > |**Ki**|ERM|1.653|0.123|0.060|0.091|
> > ||TAB|**1.418**±0.004|**0.142**±0.001|**0.116**±0.001|**0.178**±0.004|
> >
> > **Results on DrugOOD across Multiple Runs**
> >
> > |Dataset|Method|RMSE(↓)|Pearson(↑)|Tau(↑)|
> > |:-:|:-:|:-:|:-:|:-:|
> > |**Assay**|ERM|1.506|0.119|0.063|
> > ||TAB|1.537±0.008|**0.380**±0.011|**0.229**±0.001|
> > ||
> > |**Protein**|ERM|1.367|0.018|0.032|
> > ||TAB|**1.318**±0.008|**0.127**±0.016|**0.071**±0.006|
> >
> > ### In response to Weakness 3
> >
> > We fully understand the concern regarding the individual contributions of consistency and contrastive learning. In addition to the experiments presented in the last response, our ablation studies were conducted over five independent runs as well.
> >
> > We observe that TAB w/o consistency learning only slightly outperforms the ERM baseline, and this improvement is indeed not statistically significant. As discussed in the main text (lines 448–454), this phenomenon is expected: applying consistency learning alone can sometimes compress the representations too strongly, thereby weakening discriminability, while applying contrastive learning alone may amplify spurious differences without sufficient regularization.
> >
> > Importantly, when these two components are combined, they act in a complementary manner, yielding a clear performance gain. More concretely, we compute the mean performance over multiple runs for each dataset and conduct paired t-tests across datasets. The tests reveal statistically significant improvements: Pearson correlation increases with a **p-value of 0.02**, and Kendall's Tau increases with a **p-value of 0.003**. Such low p-values indicate that the improvements are unlikely to be explained by randomness or dataset variability, supporting that the full method delivers genuine and consistent benefits across datasets.
> >
> > Thus, while the contribution of each module individually may be small or variable, the joint design of TAB demonstrates a robust and significant improvement, supporting the effectiveness of integrating consistency and contrastive learning in our framework.
> >
> > |Dataset|TAB(RMSE)|TAB(Pearson)|TAB(Tau)|TAB w/o contr(RMSE)|TAB w/o contr(Pearson)|TAB w/o contr(Tau)|TAB w/o cons(RMSE)|TAB w/o cons(Pearson)|TAB w/o cons(Tau)|ERM(RMSE)|ERM(Pearson)|ERM(Tau)|
> > |:-:|:-:|:-:|:-:|:-:|:-:|:-:|:-:|:-:|:-:|:-:|:-:|:-:|
> > |I1|1.191±0.059|0.610±0.015|0.437±0.013|1.177±0.045|0.596±0.018|0.430±0.012|1.204±0.059|0.605±0.007|0.433±0.012|1.282|0.592|0.417|
> > |I2|1.421±0.052|0.613±0.013|0.427±0.008|1.546±0.052|0.603±0.016|0.409±0.031|1.498±0.057|0.570±0.018|0.404±0.012|1.672|0.533|0.393|
> > |I3|1.446±0.015|0.488±0.036|0.335±0.016|1.532±0.046|0.515±0.026|0.347±0.020|1.501±0.089|0.473±0.018|0.319±0.015|1.474|0.474|0.314|
> > |I4|1.221±0.008|0.396±0.005|0.276±0.004|1.236±0.012|0.396±0.003|0.279±0.002|1.276±0.024|0.329±0.024|0.235±0.018|1.167|0.389|0.273|
> > |I5|1.127±0.007|0.352±0.005|0.233±0.006|1.175±0.007|0.317±0.006|0.204±0.002|1.149±0.009|0.357±0.008|0.237±0.004|1.125|0.349|0.226|
> > |E1|0.808±0.030|0.226±0.025|0.156±0.036|0.829±0.060|0.215±0.018|0.135±0.014|0.920±0.041|0.203±0.002|0.145±0.004|0.855|0.200|0.139|
> > |E2|0.796±0.016|0.483±0.017|0.338±0.006|0.771±0.012|0.463±0.025|0.332±0.036|0.811±0.065|0.487±0.016|0.342±0.016|0.864|0.462|0.330|
> > |E3|1.227±0.012|0.342±0.016|0.287±0.004|1.246±0.018|0.322±0.007|0.280±0.013|1.267±0.008|0.297±0.025|0.264±0.033|1.231|0.316|0.271|
> > |Avg|1.155±0.025|0.439±0.017|0.311±0.012|1.189±0.032|0.428±0.015|0.302±0.016|1.203±0.044|0.415±0.015|0.297±0.014|1.209|0.414|0.295|

---

> > > ### Author Response · Authors · 2025-11-21
> > > **In response to Question**
> > >
> > > ### In response to Question
> > >
> > > As explained in our response to the first comment, we intentionally use the term "bioactivity" rather than "binding affinity". Bioactivity more accurately reflects the biological outcome of protein–ligand interactions, while binding affinity describes the binding strength, which does not always translate to functional biological effects. We believe this terminology fosters a more precise and biologically grounded understanding, which is especially important for AI researchers working in the life sciences domain.

---

> > > ### Comment · Reviewer_UabT · 2025-11-21
> > > **Reply to W3**
> > >
> > > I find the argument convincing. I would suggest that the authors revise the discussion and adapt claims like: "we also observe that incorporating either consistency or contrastive learning in isolation may occasionally underperform the baseline." to indicate that they have no significant effect in isolation and thus better represent the empirical results. Otherwise, I consider this concern resolved.

---

> > > > ### Author Response · Authors · 2025-11-22
> > > > **Response to Weakness 3**
> > > >
> > > > We are grateful for the suggestion. We will revise the statement to more accurately reflect the empirical findings, clarifying that incorporating either consistency or contrastive learning in isolation may occasionally risk performance degradation.
> > > >
> > > > We appreciate the reviewer's clarification and consider this concern fully addressed.

---

> ### Comment · Reviewer_UabT · 2025-11-21
> **Reply to W1**
>
> I appreciate your efforts for clarifying the terminology. First of all, I find your rationale solid and for the purposes of this paper, my concerns will be addressed if you include a short sentence summarising this argument. I think it is a valuable perspective and important to make the distinction that you make.
>
> Now, with the intention of just discussing the merits of each terminology. I personally disagree with this narrow definition of bioactivity, because although is true that the mechanism of action of most of the examples I provided can be traced back to specific drug-target interactions: in some cases they are not individual drug-target interactions, it may be dual or multi-target interactions particularly with natural antibiotics; furthermore the fact that the phenotypic responses obey to specific drug-target interactions that does not mean that we know the target for a given drug. Sometimes, it is more useful to be able to predict the phenotypic response directly rather than the biological response of a specific target. I would prefer and so I recommend the authors to try to find a different terminology that is separate both from binding affinity (which I agree just reflects the physical process) and broad phenotypic response (which is the more standard definition of bioactivity broadly). However, I reiterate that I leave this matter to the authors' discretion; I just think it will both better define the scope of the paper and prevent the broader literature to be polluted by several definitions of the same terms.

---

> > ### Comment · Reviewer_UabT · 2025-11-21
> > **Reply to W2**
> >
> > The standard deviations show that the margin of improvement is not significant. If we consider for example the average on DTGIN and calculate the lower bound of the confidence interval at 95%:
> >
> > 0.439 - 1.96 * 0.017 / sqrt(5) = 0.417; if we assume a similar dispersion for ERM, which has performance of 0.414, the difference is not statistically significant.
> >
> > The results for the other two tables do look significant. It is necessary that the authors properly revise their interpretation of the results, as claims of improvement on DTIGN are not supported by this data. Otherwise the other two sets of results do show an improvement from your method.

---

> > > ### Author Response · Authors · 2025-11-22
> > > **Response to Weakness 2**
> > >
> > > We thank the reviewer for the careful analysis. However, there is a small computation error in the provided calculation. The correct value is:
> > >
> > > $0.439 − 1.96 × 0.017 / \sqrt{5} = 0.424$
> > >
> > > not 0.417.
> > >
> > > The reviewer can see that even this lower bound (0.424) remains higher than the mean performance of ERM (0.414), indicating that the improvement is still statistically meaningful under this confidence interval.

---

> > > > ### Comment · Reviewer_UabT · 2025-11-25
> > > >
> > > > I apologize for the numerical error. But, I do not think it alters my argument. My point is that in the same way that the performance of your method has a certain dispersion and therefore a confidence interval, so does the previous method we are comparing against. Now, we don't have a measure of the dispersion of its error (and that's why all papers should publish it), so how can we account for it? You could either replicate their evaluation with as many repetitions as you have done to get dispersion (which is a computational cost, but with how narrow the margins are, needed) or we could assume that their dispersion is similar to your method. In which case your lower bound 0.424 is within their upper bound 0.414+0.014=0.428. That's why I'm arguing that it is not statistically meaningful. So, the only why that the authors could support the argument that it is indeed statistically significant would be through a replication study (which I know is an undue computational burden). However, even if you were able to demonstrate statistical significance, an increase of 0.04±0.014 is not meaningful.
> > > >
> > > > Now, the other two datasets show a more meaningful improvement with your method. So, I'm happy to update the score to 4.

---

> > > > > ### Author Response · Authors · 2025-11-27
> > > > > **Response to Weakness 2**
> > > > >
> > > > > We thank the reviewer for the careful assessment. However, we would like to clarify a common misconception underlying the comment. The reviewer concludes a lack of significance based solely on the overlap between the confidence intervals (CI) of the mean performance of the baseline and our method. This reasoning is mathematically incorrect: **overlapping CIs do not imply the absence of statistical significance** [1][2].
> > > > >
> > > > > To rigorously assess significance, we conduct a standard paired t-test by independently repeating both the baseline and our method five times. For each run, we report the mean value across all eight datasets, yielding five paired performance measurements. In addition to Pearson's $R$, we also include RMSE and Kendall's Tau for completeness, as shown in the table.
> > > > >
> > > > > |Method|RMSE(↓)|Pearson(↑)|Tau(↑)|
> > > > > |:-:|:-:|:-:|:-:|
> > > > > |ERM|1.209|0.414|0.295|
> > > > > |TAB (Ours)|**1.157**|**0.449**|**0.312**|
> > > > > ||
> > > > > |ERM|1.251|0.391|0.275|
> > > > > |TAB (Ours)|**1.162**|**0.431**|**0.303**|
> > > > > ||
> > > > > |ERM|1.241|0.394|0.282|
> > > > > |TAB (Ours)|**1.158**|**0.437**|**0.302**|
> > > > > ||
> > > > > |ERM|1.248|0.402|0.281|
> > > > > |TAB (Ours)|**1.143**|**0.434**|**0.303**|
> > > > > ||
> > > > > |ERM|1.227|0.400|0.284|
> > > > > |TAB (Ours)|**1.153**|**0.443**|**0.311**|
> > > > >
> > > > > We computed the corresponding p-values for each metric based on the five paired runs: **0.008** for RMSE, **6.25e-5** for Pearson's $R$, and **0.0004** for Kendall's Tau. By conventional statistical standards, **a result is considered significant when $p < 0.05$**. All three metrics satisfy this criterion by a substantial margin, demonstrating that the observed performance gains are statistically significant and unlikely to arise from random variation. **Thus, the CI-overlap argument cannot be used to deny the improvement of the proposed method, and the formal hypothesis tests clearly indicate that our method yields reliably better performance.**
> > > > >
> > > > > The reviewer further argues that, "even if statistically significant, an increase of 0.04 ± 0.014 is not meaningful." **We respectfully note that this judgment is inherently subjective and should be evaluated within the constraints of source-data-free setting**. Based on the measurements in the table, we first compute the mean value for each metric, and then calculate the relative improvement over the baseline, showing that **RMSE decreases by 6.5%, Pearson's $R$ increases by 9.6%, and Kendall's Tau increases by 8.0%**, indicating consistent and substantial gains across all metrics. These gains are achieved without accessing training data, constructing diverse virtual domains, or re-training the base model for dozens of epochs.
> > > > >
> > > > > Given that (1) the improvements are statistically significant, (2) the gains are consistent across all evaluation metrics, and (3) the method is lightweight and strictly test-time only, we believe the observed enhancement is both practically meaningful and highly valuable for real-world deployment scenarios.
> > > > >
> > > > > [1] Knezevic, Andrea. "Overlapping confidence intervals and statistical significance." StatNews: Cornell University Statistical Consulting Unit 73.1 (2008).
> > > > >
> > > > > [2] Payton, Mark E., Matthew H. Greenstone, and Nathaniel Schenker. "Overlapping confidence intervals or standard error intervals: what do they mean in terms of statistical significance?." Journal of insect science 3.1 (2003): 34.

---

> > > > > > ### Comment · Reviewer_UabT · 2025-11-27
> > > > > >
> > > > > > I find the arguments convincing. I appreciate the use of more rigorous testing. I am happy to raise the score to 8. My main concern was about the significance of the results both in terms of the statistical and practical meaning and the arguments in this response have resolved both concerns. Congratulations to the authors!

---

> > ### Author Response · Authors · 2025-11-22
> > **Response to Weakness 1**
> >
> > We sincerely thank the reviewer for the prompt and thoughtful feedback, and appreciate your recognition of our explanation. We fully acknowledge the reviewer's concern regarding the ambiguity and breadth of the term bioactivity. To this end, we propose to adopt "biochemical potency", which more accurately reflects the scope of our work. Biochemical potency characterizes the strength of a compound's direct interaction with a specific biological target in a cell-free biochemical assay, typically quantified by Ki, Kd, IC50, or EC50. By focusing on target-specific biochemical readouts, this terminology avoids complications related to dual or multi-target effects and does not extend to broader phenotypic activities such as antibacterial or anticancer responses. We will revise the terminology accordingly in the main manuscript.
> >
> > We are grateful for the reviewer's guidance and consider this concern fully addressed.

---

> > > ### Comment · Reviewer_UabT · 2025-11-25
> > >
> > > That's a good solution. I appreciate the effort the authors have put into resolving this minor issue.

---

### Official Review · Reviewer_oGSF · 2025-10-29

**Soundness:** 2
**Presentation:** 2
**Contribution:** 1
**Rating:** 2
**Confidence:** 5

**Summary:**

In the paper, the authors proposed methods for test-time adaptation for OOD ligand-target prediction under the assumption that  the accesses to source (training) data are prohibited. The proposed method called TAB employs data augmentation by randomly masking atoms in the binding graphs and forcing the consistency of the representation of  augmented data and the original data. Besides, contrastive learning was also proposed to enhance the representation discriminabilty and prevent feature collapse.

In the experiments, the authors studies the effectiveness of TAB against baseline approaches across DTIGN, SIU 0.6, and DrugOOD benchmark datasets. The results show that their proposed method outperforms baseline approaches with a significant margin.

**Strengths:**

The application domain is important, the problems the authors tackle is a challenging problem.

**Weaknesses:**

The assumption that source data is not available but the pretrained model is available limits the application of the work in practice. Accesses to the model especially the weights of the models usually also come with accesses to the training data as well. Making a strong assumption that is not realistic results in a small variation of  the problem  but limit its scope of application.

The methods proposed in the work such as data augmentation via randomly masking atoms, consistency and contrastive training objectives are all standard techniques.  Therefore the technical contribution of the works are limited.

Even the paper assume that there is no accesses to source data  but it also assume that there is an access to numerous test data used for augmentation and training consistency/contrastive learning.  To perform such adaptation  it requires numerous test data and the OOD setting becomes less practically interesting because in practice OOD test instances are rare.

Under OOD, the assumption that pockets are known is a strong assumption as for new target even pockets are not known in advance. The given assumption restrict the application of the work to known targets.

**Questions:**

Could you please analyse the effects of having very limitted test data say few-shot settings on the proposed methods?

---

> ### Author Response · Authors · 2025-11-21
> **Response to Weakness 1**
>
> We appreciate your valuable comments, including the recognition of the problem's significance and difficulty. We have carefully considered all feedback and present our point-by-point responses below.
>
> ### In response to Weakness 1:
>
> We would like to clarify that this scenario is both common and practically important. **It is worth noting that all three of the other reviewers explicitly affirm that our chosen application scenario is meaningful, realistic, and highly relevant in practice, which further reinforces the validity and importance of this setting.** Test-time adaptation without source data, also known as source-free domain adaptation, is already a well-established research setting, as prior work explicitly notes: "access to the source domain samples may not always be feasible in real-world applications due to different problems (e.g., storage, transmission, and privacy issues)" [1] ; "the training data in source domain required by most of the existing methods is usually unavailable in real-world applications due to privacy preserving policies" [2]. Beyond that, several earlier works also emphasize the importance and practicality of this setting [3][4][5]. Furthermore, several influential works, such as CLIP [6] and SynCLR [7], have made their pretrained model weights publicly available without releasing the corresponding training data, demonstrating that access to the model while keeping the source data private is a practical and widely adopted paradigm.
>
> In the field of drug discovery, this constraint is even more pronounced, since pharmaceutical datasets are usually proprietary and highly sensitive. Consequently, it is often necessary to decouple the final predictive asset (i.e., model weights) from the underlying intellectual property (IP) and confidential training data used to create it [8]. Federated learning represents a typical and increasingly popular practice among pharmaceutical and biomedical organizations. In such settings, end users (companies or hospitals) generally have access to the central pretrained model and its weights, which they can further fine-tune using their own private data. However, they do not have access to the original training data used to develop the central model, as it always aggregates confidential datasets contributed by multiple institutions. A real-world case is the MELLODDY Consortium [9], involving 10 major pharmaceutical companies including AstraZeneca, GSK, and Merck, where the final QSAR models can be deployed at new sites while all underlying datasets remained private, ensuring both competitive confidentiality and regulatory compliance.
>
> Thus, the "source-absent but weights-available" setting we adopt is not an artificial restriction: it aligns with established research and matches real deployment practices in pharmaceutical and biomedical applications. Our method is designed precisely for these practical scenarios, where the source data is inaccessible but adaptation to new targets or distribution shifts remains necessary.
>
> [1] VDM-DA: Virtual Domain Modeling for Source Data-free Domain Adaptation, IEEE Transactions on Circuits and Systems for Video Technology, 2021
>
> [2] Source-Free Domain Adaptation via Distribution Estimation, CVPR, 2022
>
> [3] Domain Adaptation in the Absence of Source Domain Data, WWW, 2016
>
> [4] Distant Supervised Centroid Shift: A Simple and Efficient Approach to Visual Domain Adaptation, CVPR, 2019
>
> [5] Do We Really Need to Access the Source Data? Source Hypothesis Transfer for Unsupervised Domain Adaptation, ICML, 2020
>
> [6] Learning Transferable Visual Models From Natural Language Supervision, ICML, 2021
>
> [7] Learning Vision from Models Rivals Learning Vision from Data, CVPR, 2024
>
> [8] Privacy-preserving techniques for decentralized and secure machine learning in drug discovery, Drug Discovery Today, 2023
>
> [9] MELLODDY: Cross-pharma Federated Learning at Unprecedented Scale Unlocks Benefits in QSAR without Compromising Proprietary Information, Journal of Chemical Information and Modeling, 2023

---

> > ### Author Response · Authors · 2025-11-21
> > **Response to Weakness 2 and 3**
> >
> > ### In response to Weakness 2:
> >
> > While the individual components—random atom masking, consistency regularization, and contrastive learning—are indeed established techniques, **our contribution lies in how these elements are re-purposed and integrated specifically for the source-data-free test-time adaptation setting of bioactivity prediction**.
> >
> > Our experiments demonstrate, for the first time, that the joint use of consistency regularization and contrastive objectives is essential for addressing molecular graph data and the unique challenges of bioactivity prediction in the source-data-free test-time adaptation setting. As shown in our ablation study (Section 4.4.1), consistency-only training or contrastive-only training can sometimes harm performance. Only when the two objectives are combined do we observe stable and significant improvements, which is a non-trivial finding that highlights the task-specific design needed for test-time adaptation on molecular data.
> >
> > Furthermore, we provide a principled explanation for why the proposed combination is effective, and we validate this explanation through case studies. Our atom-masking augmentation perturbs atom features while preserving 3D coordinates; this intentionally disrupts local chemical motifs, weakens shortcut patterns, and forces the model to rely on more informative geometric cues during adaptation. When consistency and contrastive objectives are applied on these geometry-preserving perturbations, the model is encouraged to extract representations that are both stable and discriminative with respect to biologically meaningful spatial interactions. As shown in our case analyses (Section 4.4.2), when removing key binding atoms—which consequently disrupts the essential geometric cues, the model’s sensitivity to these perturbations becomes evident. Specifically, this manipulation yields an additional decrease in $\Delta\hat{y}$ of 0.479, 0.027, and 1.369 on the DTIGN I1, I2, and I3 datasets, respectively. Such a consistent drop indicates that our method indeed focuses more on meaningful binding geometry.
> >
> > Overall, although the individual components are standard, their interplay and demonstrated necessity in the source-data-free test-time adaptation setting of bioactivity prediction constitute a substantive technical contribution, along with the empirical and mechanistic insights revealed through our study.
> >
> > ### In response to Weakness 3:
> >
> > **In practical bioactivity prediction and virtual screening scenarios, there is typically a large number of test compounds**, which motivates the use of deep learning models to efficiently prioritize candidates for experimental validation. For instance, Atomwise employs its AtomNet platform to virtually screen **over 16 billion compounds** for potential protein-drug interactions [1]. Recursion has predicted protein–compound interactions across approximately **36 billion** molecules from the Enamine REAL Space chemical library [2]. The Enamine REAL collection itself comprises **over 15.5 billion** make-on-demand compounds [3], and its practical value has been demonstrated by the discovery of highly potent AmpC β-lactamase inhibitors, D4 dopamine receptor ligands, and Kelch-like ECH-associated protein 1 (KEAP1) inhibitors, with these successes reported in Nature [4][5].
> > In contrast, when only a few test instances are available, it is often more practical to evaluate them directly in wet-lab experiments. Accordingly, our method is designed for realistic, large-scale screening settings.
> >
> > [1] Hale, Conor. "Atomwise Raises $123M to Expand AI-Powered Drug Design Efforts." Fierce Biotech, 14 Aug. 2020.
> >
> > [2] "A Deep Dive into Screening 36 Billion Compounds: Q&A with Stephen MacKinnon." Recursion, 8 Aug. 2023.
> >
> > [3] Grygorenko, Oleksandr. "REAL Space rapidly expands." Enamine, 6 Nov. 2020.
> >
> > [4] Ultra-large library docking for discovering new chemotypes. Nature, 2019.
> >
> > [5] An open-source drug discovery platform enables ultra-large virtual screens. Nature, 2020.

---

> > > ### Author Response · Authors · 2025-11-21
> > > **Response to Weakness 4 and Question**
> > >
> > > ### In response to Weakness 4:
> > >
> > > We apologize for any wording that may have unintentionally suggested that our method relies on *a priori* knowledge of the true binding pocket. In practice, pockets can be obtained through two standard and widely used procedures:
> > >
> > > **1. From experimental protein-ligand complex structures.**
> > >
> > > When crystallized complexes are available, the binding pocket can be directly identified as the set of residues surrounding the bound ligand. For structure-based datasets such as **DTIGN** and **SIU**, these pocket definitions are already provided as part of the dataset construction, so no additional assumptions are introduced by our method.
> > >
> > > **2. From computational pocket detection models.**
> > >
> > > For targets without known ligand-bound structures, pocket-like regions can be predicted using established tools. For example, in **DrugOOD**, where only protein sequences are available, we use **Fpocket** [1] to automatically detect candidate pockets. Fpocket identifies geometric cavities and ranks them by pocketness scores, providing plausible binding pockets even for unseen targets.
> > >
> > > Therefore, our method **does not require pockets to be pre-defined or known**. Instead, pockets can be (i) extracted directly when protein–ligand complexes are available, or (ii) predicted using well-established pocket detection algorithms when only protein sequences exist. This flexibility ensures that our approach remains applicable to **new, unseen targets** and does not limit its usability to known targets.
> > >
> > > [1] Fpocket: an open source platform for ligand pocket detection, BMC bioinformatics, 2009
> > >
> > > ### In response to Question
> > >
> > > We would like to clarify again that our method is primarily designed for realistic, large-scale screening scenarios where a substantial number of test compounds are available. In such settings, deep learning models can efficiently prioritize candidates for experimental validation. When only a handful of compounds are available, it is often more practical to evaluate them directly in wet-lab experiments.

---

> ### Author Response · Authors · 2025-11-27
> **Follow-up on Reviewer Comments**
>
> We hope the above clarifications have adequately addressed your concerns. If you are satisfied, we would be grateful if you could consider updating your evaluation score to reflect the discussion. We remain fully committed to addressing any remaining questions or points you may have during the discussion phase.

---

### Official Review · Reviewer_Weoy · 2025-10-30

**Soundness:** 2
**Presentation:** 4
**Contribution:** 3
**Rating:** 6
**Confidence:** 5

**Summary:**

This paper tackles the important problem of out-of-domain (OOD) generalization in bioactivity prediction under a source data-absent setting. The authors propose TAB, a test-time adaptation framework combining (1) uncertainty-weighted consistency learning and (2) contrastive optimization. Extensive experiments on DTIGN, SIU 0.6, and DrugOOD show consistent gains over a variety of state-of-the-art OOD generalization baselines.

**Strengths:**

1.The writing is clear and figures are well-designed, facilitating understanding.

2.The chosen application scenario is highly relevant and meaningful.

3.The experimental results are strong and demonstrate the effectiveness of the proposed method

**Weaknesses:**

1.Realistic source data-absent scenario. The proposed source-absent setting assumes that the training data are inaccessible. While this may occur in industrial contexts, for example when dealing with large-scale proprietary pre-trained models, the experiments in the paper mainly use a small DTIGN model trained on publicly available ChEMBL data. It is unclear whether TAB’s effectiveness in true source-absent scenarios can be reliably inferred. Evaluating TAB on larger pre-trained models would strengthen the paper.

2.Ablation studies are insufficient. The paper does not provide ablations for the update strategy or the newly proposed masking strategy. Including these analyses would clarify the contributions of each component.

3.Batch update and sequential sensitivity. TAB accumulates model updates using EMA and memory to reduce sensitivity. An ablation study examining the effects of batch size and test sample order would help assess robustness and stability.

4.Clarification on base model usage across datasets. It is unclear whether the same pre-trained DTIGN model is applied to all three datasets or if models are re-trained for each dataset. If the former, the authors should clarify potential overlaps or differences between training and test distributions, particularly for the latter two datasets.

5.Comparison baselines. The reported comparisons are all against source-available methods. It would be useful to include comparisons with source-absent baselines or other approaches designed for realistic deployment constraints.

**Questions:**

please respond to the weakness

---

> ### Author Response · Authors · 2025-11-21
> **Response to Weakness 1 and 2**
>
> We sincerely thank you for the encouraging feedback and for appreciating both the real-world relevance of our scenario and the strength of our experimental results. Below, we provide detailed responses to your comments.
>
> ### In response to Weakness 1
>
> We evaluate our method on a substantially larger bioactivity-pretrained model. MBP is a recent pretrained model based on ChEMBL-Dock, which has more than 300k experimentally measured bioactivity labels and about 2.8M docked 3D structures [1]. The results in the table below show that our method remains effective when applied to MBP, thereby strengthening its relevance to realistic source-absent scenarios.
>
> |Dataset|Method|RMSE(↓)|Pearson(↑)|Tau(↑)|
> |:-:|:-:|:-:|:-:|:-:|
> |**I1**|MBP ERM|1.661|0.044|0.040|
> ||MBP TAB|1.741|**0.056**|**0.057**|
> ||
> |**I2**|MBP ERM|1.669|-0.118|-0.067|
> ||MBP TAB|**1.623**|**0.131**|**0.039**|
> ||
> |**I3**|MBP ERM|2.437|-0.062|-0.011|
> ||MBP TAB|**2.345**|**0.085**|**0.067**|
>
> [1] Multi-task bioassay pre-training for protein-ligand binding affinity prediction, Briefings in Bioinformatics, 2024.
>
> ### In response to Weakness 2
>
> We have added ablation studies for both the update strategy (EMA) and the newly introduced masking strategy. The results are summarized in the table below. *TAB w/o EMA* indicates removing the EMA update strategy; *TAB (substitution)* replaces atom masking with random atom substitution; *TAB (coord. jitter)* replaces atom masking with perturbations applied to atomic 3D coordinates; and *TAB (mask)* corresponds to our original implementation.
>
> |Dataset|Method|RMSE(↓)|Pearson(↑)|Tau(↑)|
> |:-:|:-|:-:|:-:|:-:|
> ||ERM|1.282|0.592|0.417|
> ||TAB w/o EMA|1.272|0.572|0.418|
> |**I1**|TAB (substitution)|1.232|0.592|0.412|
> ||TAB (coord. jitter)|1.470|0.503|0.347|
> ||TAB (mask)*|1.180|0.614|0.427|
> ||
> ||ERM|1.672|0.533|0.393|
> ||TAB w/o EMA|1.585|0.556|0.404|
> |**I2**|TAB (substitution)|1.513|0.581|0.416|
> ||TAB (coord. jitter)|1.665|0.543|0.398|
> ||TAB (mask)*|1.481|0.605|0.419|
> ||
> ||ERM|1.474|0.474|0.314|
> ||TAB w/o EMA|1.467|0.474|0.323|
> |**I3**|TAB (substitution)|1.449|0.477|0.319|
> ||TAB (coord. jitter)|1.493|0.446|0.299|
> ||TAB (mask)*|1.433|0.537|0.356|
>
> *\*Notes: TAB (mask) represents our original implementation.*
>
> **Update Strategy (EMA)**
>
> We observe that removing EMA leads to substantially smaller performance improvement. This indicates that EMA plays a critical role in stabilizing the test-time optimization, which prevents the model from drifting toward noisy or unstable updates.
>
> **Masking Strategy**
>
> We also analyze these two alternative augmentation methods compared to our masking strategy. Random atom substitution can, similar to masking, disrupt biased substructures and encourage the model to rely on geometry-derived cues that are more relevant to binding. While this augmentation provides some benefit, atom substitution may generate chemically implausible molecules and introduce additional noise, resulting in weaker gains compared to masking atom features. In contrast, perturbing 3D coordinates significantly undermines the ability of the model to learn accurate geometric information, which is essential for ligand–target binding. This leads to unstable updates and, in some cases, even worse performance than the pretrained baseline.

---

> > ### Author Response · Authors · 2025-11-21
> > **Response to Weakness 3, 4 and 5**
> >
> > ### In response to Weakness 3
> >
> > We have further added ablation studies examining the effects of batch size and test sample order, and the results are summarized in the following tables.
> >
> > **Effect of Batch Size**
> > |Dataset|Method|Batch Size|RMSE(↓)|Pearson(↑)|Tau(↑)|
> > |:-:|:-:|:-:|:-:|:-:|:-:|
> > ||ERM|-|1.282|0.592|0.417|
> > ||TAB|4|1.186|0.604|0.442|
> > |**I1**|TAB|8|1.166|0.627|0.448|
> > ||TAB|16|1.180|0.614|0.427|
> > ||TAB|32|1.272|0.601|0.431|
> > ||
> > ||ERM|-|1.672|0.533|0.393|
> > ||TAB|4|1.641|0.550|0.392|
> > |**I2**|TAB|8|1.453|0.581|0.409|
> > ||TAB|16|1.481|0.605|0.419|
> > ||TAB|32|1.508|0.566|0.396|
> > ||
> > ||ERM|-|1.474|0.474|0.314|
> > ||TAB|4|1.443|0.487|0.328|
> > |**I3**|TAB|8|1.438|0.483|0.334|
> > ||TAB|16|1.433|0.537|0.356|
> > ||TAB|32|1.446|0.490|0.400|
> >
> > We evaluate four batch sizes (4, 8, 16, 32). All configurations outperform the pretrained baseline, demonstrating that TAB is robust across a wide range of batch settings. Besides, we observe a “rise-then-fall” trend as batch size increases: moderate batch sizes yield the best performance, while extremely large batches lead to slightly reduced gains.
> >
> > **Effect of Test Sample Order**
> > |Dataset|Method|Order|Batch Size|RMSE(↓)|Pearson(↑)|Tau(↑)|
> > |:-:|:-:|:-:|:-:|:-:|:-:|:-:|
> > ||ERM|-|-|1.282|0.592|0.417|
> > |**I1**|TAB|Fixed|16|1.187|0.600|0.439|
> > ||TAB|Shuffle|16|1.180|0.614|0.427|
> > ||
> > ||ERM|-|-|1.672|0.533|0.393|
> > |**I2**|TAB|Fixed|16|1.647|0.571|0.411|
> > ||TAB|Shuffle|16|1.481|0.605|0.419|
> > ||
> > ||ERM|-|-|1.474|0.474|0.314|
> > |**I3**|TAB|Fixed|16|1.440|0.482|0.334|
> > ||TAB|Shuffle|16|1.433|0.537|0.356|
> >
> > Our default setting uses randomly shuffled test samples, and we compare it with a fixed sequential order. The shuffled order consistently performs better. This is likely because our test dataset is relatively large and contains samples from diverse latent sub-distributions. Random shuffling ensures that each mini-batch contains a heterogeneous mixture of samples, preventing the model from overfitting to any local region of the test distribution. In contrast, a fixed order may present temporally or structurally correlated samples consecutively, which can lead to biased updates and less stable adaptation.
> >
> > ### In response to Weakness 4
> >
> > The DTIGN model is independently re-trained on each dataset. Since every dataset specifies its own OOD configuration, the associated training and test distributions are intrinsically well-defined and explicitly non-overlapping.
> >
> > ### In response to Weakness 5
> >
> > To the best of our knowledge, there are currently no existing source-absent baselines in the field of bioactivity prediction. All prior domain adaptation methods in this domain rely on access to source data during adaptation. To the best of our knowledge, our work is the first to propose and systematically investigate the source-data-free test-time adaptation setting for this task. Consequently, all comparisons are against source-available methods, which represents the current state-of-the-art baselines.

---

> ### Author Response · Authors · 2025-11-27
> **Follow-up on Reviewer Comments**
>
> We greatly appreciate the time and effort you have devoted to reviewing our submission. As the discussion period draws to a close, we would like to confirm whether our responses have fully addressed your comments. If any points remain unclear or require additional explanation, we would be happy to provide further details. Otherwise, we would be grateful if you could take our clarifications into consideration when revising your evaluation score. We sincerely welcome any feedback you may have.

---

### Official Review · Reviewer_1xoV · 2025-11-01

**Soundness:** 2
**Presentation:** 3
**Contribution:** 3
**Rating:** 6
**Confidence:** 3

**Summary:**

This paper introduces TAB, a test-time adaptation framework to improve out-of-domain (OOD) bioactivity prediction in scenarios where the original source data is inaccessible. The proposed method uses an uncertainty-weighted consistency strategy and contrastive learning to adapt a pre-trained model, encouraging it to focus on important binding regions while ignoring spurious substructures. Experiments across multiple benchmarks demonstrate that this source-data-absent approach significantly outperforms existing methods in scaffold, protein, and assay-based OOD settings.

**Strengths:**

1. This work develops the first Test-Time Adaptation (TTA) method tailored to bioactivity prediction, presenting a novel and practically relevant problem formulation.

2. The proposed framework performs self-supervised optimization during testing without requiring complex architectural changes or extensive additional labeled data, making the overall approach simple and easy to follow.

3. The evaluation on three OOD benchmarks with different focuses covers various distribution shift scenarios, demonstrating strong generalizability and robustness.

**Weaknesses:**

1. The loss weights for consistency learning ($\alpha$) and contrastive learning ($\beta$) are fixed at 1.0 across all experiments. It would be helpful to include sensitivity analysis for these key hyperparameters to validate model robustness.

2. The existing ablation study (Table 5) validates the necessity of consistency learning and contrastive learning modules. Could authors further isolate uncertainty weighting for separate validation, as this appears to be an important contribution?

3. How is the pre-trained encoder in Fig. 2 or Section 3.2 obtained?

4. Minor:
(a) Typos in Equation 3;
(b) Suggest adding an analysis of TAB's efficiency and computational resource consumption.

**Questions:**

See Weaknesses

---

> ### Author Response · Authors · 2025-11-21
>
> We sincerely thank the reviewer for the encouraging comments and for recognizing the novelty and practical relevance of our setting, as well as the robustness demonstrated across diverse OOD benchmarks. Here are our responses to your concerns.
>
> ### In response to Weakness 1
>
> We conduct a sensitivity analysis on the consistency weight ($\alpha$) and contrastive weight ($\beta$), covering a broad range of settings while keeping $\alpha$ + $\beta$ = 2 for comparability, as shown below.
>
> **Different configurations of the weight of consistency and contrastive learning**
>
> |Dataset|Method|$\alpha$|$\beta$|RMSE↓|Pearson↑|Tau↑|
> |:-:|:-:|:-:|:-:|:-:|:-:|:-:|
> ||ERM|-|-|1.282|0.592|0.417|
> ||TAB|0.2|1.8|1.265|0.590|0.424|
> ||TAB|0.6|1.4|1.374|0.602|0.424|
> |**I1**|TAB|1|1|1.180|0.614|0.427|
> ||TAB|1.4|0.6|1.165|0.614|0.417|
> ||TAB|1.8|0.2|1.289|0.605|0.426|
> ||
> ||ERM|-|-|1.672|0.533|0.393|
> ||TAB|0.2|1.8|1.460|0.603|0.420|
> ||TAB|0.6|1.4|1.453|0.601|0.427|
> |**I2**|TAB|1|1|1.481|0.605|0.419|
> ||TAB|1.4|0.6|1.408|0.597|0.439|
> ||TAB|1.8|0.2|1.427|0.599|0.411|
> ||
> ||ERM|-|-|1.474|0.474|0.314|
> ||TAB|0.2|1.8|1.448|0.463|0.327|
> ||TAB|0.6|1.4|1.458|0.458|0.319|
> |**I3**|TAB|1|1|1.433|0.537|0.356|
> ||TAB|1.4|0.6|1.552|0.498|0.340|
> ||TAB|1.8|0.2|1.588|0.510|0.347|
>
> We can observe that performance remains stable across a wide range of α/β ratios, indicating that our method is not overly sensitive to the exact choice of these hyperparameters. Besides, the default setting ($\alpha$ = 1, $\beta$ = 1) consistently yields competitive performance, supporting our choice of using equal weighting in the main experiments.
>
> ### In response to Weakness 2 ###
>
> We conduct an additional ablation where the uncertainty-weighted loss is replaced with uniform weighting (TAB *w/o uw*). As shown below, removing uncertainty weighting consistently decreases performance.
>
> |Dataset|Method|RMSE(↓)|Pearson(↑)|Tau(↑)|
> |:-:|:-:|:-:|:-:|:-:|
> ||ERM|1.282|0.592|0.417|
> |**I1**|TAB *w/o uw*|1.232|0.591|0.422|
> ||TAB|1.180|0.614|0.427|
> ||
> ||ERM|1.672|0.533|0.393|
> |**I2**|TAB *w/o uw*|1.609|0.576|0.416|
> ||TAB|1.481|0.605|0.419|
> ||
> ||ERM|1.474|0.474|0.314|
> |**I3**|TAB *w/o uw*|1.435|0.484|0.336|
> ||TAB|1.433|0.537|0.356|
>
> These results confirm that uncertainty-aware weighting plays a crucial role, as it adaptively down-weights unreliable predictions during test-time optimization, reducing the risk of propagating noisy updates.
>
> ### In response to Weakness 3 ###
>
> For each benchmark dataset, we adopt its official train/test split and train a base encoder on the corresponding training set, which we refer to as the pre-trained encoder. Importantly, this setup merely serves to instantiate the test-time adaptation framework rather than impose any dependence on training data. In realistic source-free scenarios, the encoder can indeed be replaced by any off-the-shelf pre-trained model.
>
> ### In response to Weakness 4 ###
>
> Thanks for your careful reading and valuable suggestions.
>
> For (a), we have corrected the typo in Equation 3:
> $$\mathcal{L}\_{cons}=\frac{1}{B}\sum\_{i=1}^{B}{w\_i \cdot \left(1-\frac{\mathbf{f}\_{i}^{o}\cdot\mathbf{f}\_{i}^{a}}{\left\||\mathbf{f}\_{i}^{o}\right\||\left\||\mathbf{f}\_{i}^{a}\right\||}\right)}$$
>
> For (b), our TAB framework is indeed highly efficient in computation. Unlike prior domain adaptation methods that rely on training data and require generating diverse virtual domains and re-training the base model for dozens of epochs [1][2][3][4], TAB operates solely on the target test set without accessing any source data. The adaptation is lightweight, and does not require re-training the base encoder or synthesizing additional data.
>
> To quantify this advantage, we report the per-epoch running time on the DTIGN I3 dataset:
> - SRGNN [1], 352 s/epoch
> - EERM [2], 563 s/epoch
> - CIA  [3], 602 s/epoch
> - CaNet [4], 869 s/epoch
> - **Ours (TAB), 147 s/epoch**
>
> Moreover, while these methods typically require dozens of epochs (often 50–200 epochs depending on the methods and dataset) to complete re-training, TAB completes adaptation in only 15 epochs, reinforcing its clear computational superiority.
>
> [1] Overcoming the Limitations of Localized Graph Training Data, NeurIPS 2021.
>
> [2] Handling Distribution Shifts on Graphs: An Invariance Perspective, ICLR 2022.
>
> [3] Dissecting the Failure of Invariant Learning on Graphs, NeurIPS 2024.
>
> [4] Graph Out-of-Distribution Generalization via Causal Intervention, WWW 2024.

---

> > ### Author Response · Authors · 2025-11-27
> > **Follow-up on Reviewer Comments**
> >
> > Thank you once again for your valuable comments on our submission. As the discussion phase is approaching its end, we would like to kindly confirm whether we have sufficiently addressed all of your concerns. Should there be any remaining questions or areas requiring further clarification, please do not hesitate to let us know. If you are satisfied with our responses, we would greatly appreciate your consideration in adjusting the evaluation scores accordingly. We sincerely look forward to your feedback.

---

### Author Response · Authors · 2025-12-01
**Author Final Remarks**

We thank all reviewers for their insightful feedback and sincerely appreciate the Area Chair’s thoughtful assessment and effort in making an informed final decision.

In this paper, **we present the first exploration of a more realistic setting for bioactivity prediction, where models are expected to adapt to out-of-domain (OOD) distributions without access to source data**, reflecting the proprietary nature of pharmaceutical datasets. To tackle this problem, **we propose a test-time adaptation framework tailored to bioactivity prediction that adapts models to unseen OOD distributions using only unlabeled target data**. Extensive experiments across DTIGN, SIU, and DrugOOD show that **our method achieves substantial performance gains under multiple OOD scenarios, including scaffold-, protein-, and assay-based shifts**. On SIU-0.6 Kd, our method improves Pearson's $R$, Kendall's $\tau$, and Spearman's $\rho$ by 22.8%, 31.6%, and 31.3%, respectively, while on DrugOOD-Assay it achieves substantial absolute gains of 0.269 in Pearson's $R$ and 0.167 in Kendall's $\tau$. Notably, **our framework consistently outperforms state-of-the-art baselines that do rely on source data, highlighting its effectiveness and practical relevance in source-data-free scenarios**.

**Before the reviewer response phase closed, our submission received scores of 6, 6, 2 and 8, resulting in an average score of 5.5. We would like to highlight that Reviewer UabT raised the score from 2 to 4 and subsequently to 8, while Reviewer 1xoV and Weoy assigned scores of 6 and 6 but did not give any feedback before the end of reviewer response period**. We believe our rebuttal has thoroughly addressed all concerns raised by the reviewers.

We are encouraged that all three reviewers (1xoV, Weoy, and UabT) recognize not only the value and realism of the proposed source-data-free setting, but also the strength of our experimental results. They explicitly state that:
- "This work develops the first Test-Time Adaptation (TTA) method tailored to bioactivity prediction, **presenting a novel and practically relevant problem formulation**."
- "The evaluation on three OOD benchmarks with different focuses **covers various distribution shift scenarios, demonstrating strong generalizability and robustness**."
- "**The chosen application scenario is highly relevant and meaningful**."
- "**The experimental results are strong and demonstrate the effectiveness of the proposed method**."

During the rebuttal period, we perform additional experiments and provide extensive analyses addressing every concern raised by Reviewers 1xoV and Weoy—including loss weighting, uncertainty weighting, method efficiency, scaling to larger pre-trained models, update strategy, masking strategy, and the effects of batch size and test sample order. For each question, we provide point-by-point clarifications and new empirical evidence, which we believe fully resolve their concerns.

We are especially grateful to Reviewer UabT for the rigor and promptness of the discussion. We clarify the distinction between "bioactivity" and "binding affinity" and demonstrate the statistical significance of our improvements using *paired t-tests*. Through multiple rounds of in-depth exchanges, we are ultimately able to address the reviewer's concerns and reach a consensus. The reviewer states that:
- "**I appreciate your efforts for clarifying the terminology**. First of all, **I find your rationale solid**..."
- "**I find the arguments convincing. I appreciate the use of more rigorous testing. I am happy to raise the score to 8. My main concern was about the significance of the results both in terms of the statistical and practical meaning and the arguments in this response have resolved both concerns. Congratulations to the authors!**".

We sincerely appreciate the reviewer's careful re-evaluation and the substantial score improvements from 2 → 4 → 8.

In response to Reviewer oGSF's concerns about the applicability and assumptions of our setting, we first provide extensive references showing that our application scenario is realistic and widely relevant in practical drug discovery, **a view also supported by the other three reviewers**. We further clarify that real-world bioactivity prediction and virtual screening typically involve large numbers of test compounds aimed at efficient pre-screening; thus, assuming a very limited few-shot test setting would not be realistic. Finally, we address the misunderstanding regarding *a priori* knowledge of the binding pocket: in practice, binding pockets can be reliably obtained from experimental protein-ligand complex structures or established computational pocket detection models. Taken together, we believe these clarifications adequately resolve the reviewer's concerns.

In closing, we would like to once again express our sincere gratitude to all reviewers and the Area Chair for their time, effort, and constructive feedback throughout the review process.

---

### Meta-Review · Area_Chair_PSBx · 2026-01-06

**Summary:**

The main reviewer concern is the lack of ablation studies and the application of the method in realistic source data-absent scenario.

**Reviewer Concerns:**

The reviewer concerns have been addressed by the authors. And some reviewers have increased their score from 2 to 8.

**Reviewer Scores:**

Many reviewers have participated in the discussion and changed their score.

---

### Decision · Program_Chairs · 2026-01-26

Accept (Poster)